# ZMYM2 controls human transposable element transcription through distinct co-regulatory complexes

**Danielle J Owen**[†‡], **Elisa Aguilar-Martinez**[†‡], **Zongling Ji**[†‡], **Yaoyong Li**[†‡], **Andrew D Sharrocks***

Faculty of Biology, Medicine and Health, University of Manchester, Michael Smith Building, Oxford Road, Manchester, United Kingdom

**\*For correspondence:**
andrew.d.sharrocks@manchester.ac.uk

[†]These authors contributed equally to this work

[‡]Joint first authors

**Competing interest:** The authors declare that no competing interests exist.

**Abstract** ZMYM2 is a zinc finger transcriptional regulator that plays a key role in promoting and maintaining cell identity. It has been implicated in several diseases such as congenital anomalies of the kidney where its activity is diminished and cancer where it participates in oncogenic fusion protein events. ZMYM2 is thought to function through promoting transcriptional repression and here we provide more evidence to support this designation. Here we studied ZMYM2 function in human cells and demonstrate that ZMYM2 is part of distinct chromatin-bound complexes including the established LSD1-CoREST-HDAC1 corepressor complex. We also identify new functional and physical interactions with ADNP and TRIM28/KAP1. The ZMYM2-TRIM28 complex forms in a SUMO-dependent manner and is associated with repressive chromatin. ZMYM2 and TRIM28 show strong functional similarity and co-regulate a large number of genes. However, there are no strong links between ZMYM2-TRIM28 binding events and nearby individual gene regulation. Instead, ZMYM2-TRIM28 appears to regulate genes in a more regionally defined manner within TADs where it can directly regulate co-associated retrotransposon expression. We find that different types of ZMYM2 binding complex associate with and regulate distinct subclasses of retrotransposons, with ZMYM2-ADNP complexes at SINEs and ZMYM2-TRIM28 complexes at LTR elements. We propose a model whereby ZMYM2 acts directly through retrotransposon regulation, which may then potentially affect the local chromatin environment and associated coding gene expression.

## eLife assessment

ZMYM2 is a transcriptional corepressor but little was known about how it is recruited to chromatin. This **important** study reveals that ZMYM2 homes to distinct classes of retrotransposons bound by the TRIM28 and ChAHP complexes in human cells, which is broadly relevant for the field of transcriptional regulation. Much of the evidence supporting the claims of the authors is **convincing**. Since widespread ZMYM2-mediated control of transposon activity is not apparent in RNA-seq data, further experiments are needed to demonstrate a more general role beyond the retrotransposons analysed in this study.

## Introduction

The zinc finger protein ZMYM2 (otherwise known as ZNF198) was originally identified as part of an oncogenic fusion protein with the receptor tyrosine kinase FGFR1 in myeloproliferative disease (*Xiao et al., 1998*; *Reiter et al., 1998*). In this context, the N-terminal portion of ZMYM2 promotes the oligomerisation and activation of FGFR1 (*Xiao et al., 2000*). Further disease links have recently been uncovered where heterozygous loss of function ZMYM2 mutations lead to congenital anomalies of

the kidney and urinary tract (CAKUT) and chronic kidney disease (*Connaughton et al., 2020*). The latter observation suggests a potential developmental role and ZMYM2 has been shown to play important roles in the early stages of embryonic stem cell (ESC) differentiation and commitment. ZMYM2 promotes the transition from totipotency to pluripotency in mice (*Yang et al., 2020*). Similarly, in human cells, ZMYM2 plays a key role in determining ESC identity by allowing the transition to and maintenance of the primed pluripotent state (*Lezmi et al., 2020*). Furthermore, ZMYM2 has been shown to play a key role in opposing reprogramming in human fibroblasts where ZMYM2 loss increases reprogramming efficiency (*Toh et al., 2016*; *Lawrence et al., 2019*).

Molecularly, ZMYM2 has been shown to act as a transcriptional repressor and can bind to the LSD1-CoREST-HDAC1 corepressor complex (LCH) via its zinc fingers (*Gocke and Yu, 2008*). It is through this repressive complex that ZMYM2 promotes pluripotency in mice (*Yang et al., 2020*). However, ZMYM2 has been shown to have an increasingly complex set of interaction partners such as the transcription factors TBX18 with a potential role in ureter development (*Lüdtke et al., 2022*) and B-MYB and hence a potential link to cell cycle control (*Cibis et al., 2020*). Proteomic screens have also revealed multiple ZMYM2 binding partners, either from using ZMYM2 itself as a bait (*Connaughton et al., 2020*) or through detection as a component of other repressive complexes by using baits such as LSD1/KDM1A and HDAC2 (*Hakimi et al., 2003*; *Shi et al., 2005*). The range of interaction partners solidifies ZMYM2 as a potential transcriptional repressor protein, but additional roles are hinted at such as DNA repair, exemplified by its ability to antagonise 53BP1 function at DNA double strand breaks to favour repair by homologous recombination (*Lee et al., 2022*).

We identified ZMYM2 in a screen for multi-SUMO binding proteins (*Aguilar-Martinez et al., 2015*) which builds on the complex interplay previously observed between ZMYM2 and SUMO. Additional work supports the non-covalent SUMO binding activity of ZMYM2 (*Guzzo et al., 2014*) although the two studies differed in mapping of the SUMO interacting motifs, with the former mapping them to the N-terminal part of the protein and the latter mapping them to the centrally located zinc finger region. Furthermore, ZMYM2 itself is covalently modified with SUMO (*Kunapuli et al., 2006*) suggesting a complex and important role for SUMO in determining ZMYM2 activity. Indeed, SUMO binding is needed for recruitment of ZMYM2 to chromatin (*Aguilar-Martinez et al., 2015*) and the ability of ZMYM2 to recruit SUMOylated HDAC-1 (*Gocke and Yu, 2008*).

In this study, we further investigated the molecular activities of ZMYM2 and demonstrated that it is associated with two distinct chromatin-bound complexes containing either ADNP or TRIM28. Both types of ZMYM2 chromatin binding regions are associated with different subclasses of retrotransposons. We focussed on TRIM28 and demonstrated widespread functional cooperativity with ZMYM2 in gene regulation. However, ZMYM2-TRIM28 binding regions are directly associated with ERV retrotransposable elements rather than with protein coding genes, suggesting an indirect mechanism for transcriptional control of the coding genome.

## Results

### ADNP is a coregulatory binding partner for ZMYM2

To investigate the molecular function of ZMYM2, we used the mass spectrometry approach RIME (*Mohammed et al., 2013*) to identify proteins co-binding with ZMYM2 on chromatin. We performed several independent experiments using an anti-flag antibody for immunoprecipitation of ZMYM2 from a U2OS cell line containing a Flag-tagged version of ZMYM2 (*Aguilar-Martinez et al., 2015*) or using an antibody against endogenous ZMYM2 in U2OS cells. We identified 27 proteins in all three experiments with an average of >4 spectral counts (*Supplementary file 3*). Among these, the highest scoring interactors were the transcriptional co-regulators TRIM28 and ADNP (*Figure 1A*). ZMYM2 and ADNP are part of a larger interacting network of proteins identified in the RIME analysis (*Figure 1B*), which includes the other two components of the ChAHP complex, CHD4 and CBX1/HP1β (*Kaaij et al., 2019*). Another prominent complex identified was the GTF3C complex (*Figure 1—figure supplement 1A*) which is consistent with a previous study that identified ZMYM2 from reciprocal RIME experiment using GTF3C2 as bait (*Ferrari et al., 2020*). A large proportion of the interactors were previously identified in two proteomic screens for ZMYM2 binding proteins from mouse ESCs (*Yang et al., 2020*) and/or human HEK293 cells (*Connaughton et al., 2020*). They are also often found as binding partners in proteomic screens for other chromatin regulators as exemplified by BEND3,

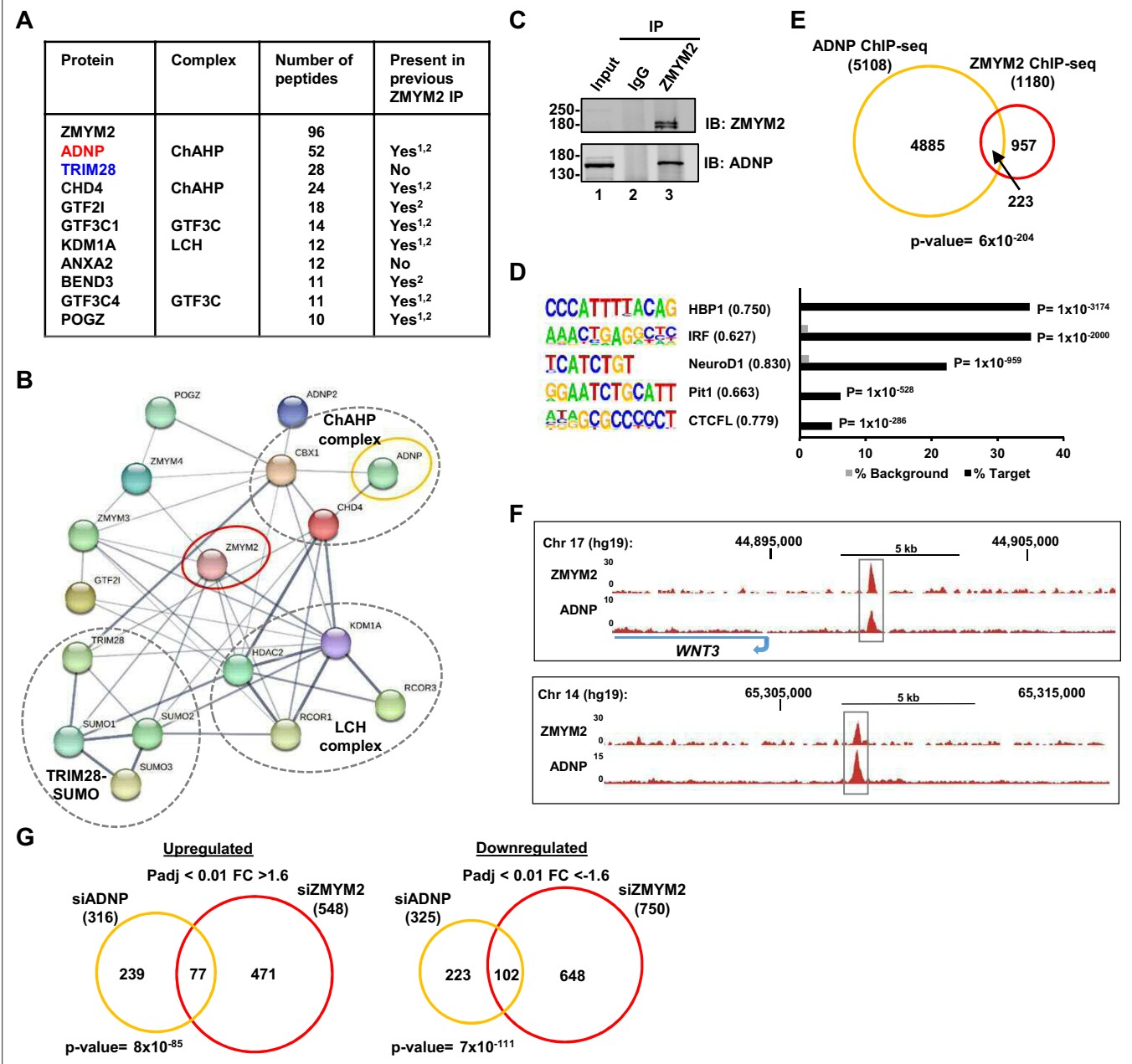

**Figure 1.** RIME analysis identifies ADNP as a co-regulatory partner of ZMYM2. (**A**) Summary table of the ten top scoring interactors for ZMYM2. The average number of peptides across three RIME experiments are shown and whether detected previously in ZMYM2 IP-mass spectrometry experiments is indicated (*Yang et al., 2020*; *Connaughton et al., 2020*). Core members of the ChAHP and GTF3C complexes are highlighted. (**B**) Depiction of interactions between ZMYM2 binding partners found in RIME experiments with known previous molecular interactions found in the STRING database (*Jensen et al., 2009*). (**C**) Co-immunoprecipitation analysis of ADNP with ZMYM2. Immunoprecipitation (IP) was performed with ZMYM2 or control IgG antibody from U2OS cells and resulting proteins detected by immunoblotting (IB) with the indicated antibodies. Molecular weight markers (kDa) and 10% input are shown. (**D**) De novo motif analysis of ADNP binding regions. The top five most significantly enriched motifs are shown, along with motif similarity to the indicated protein in brackets. (**E**) Venn diagram showing the overlap between ZMYM2 and ADNP binding regions. (**F**) UCSC genome browser of ChIP-seq data on example genomic loci showing binding of both ZMYM2 and ADNP. (**G**) Venn diagram showing overlaps in genes upregulated (left) or downregulated (right) following ADNP (orange) or ZMYM2 (red) depletion (fold change >1.6; Padj <0.01). See also *Figure 1—figure supplement 1*.

The online version of this article includes the following source data and figure supplement(s) for figure 1:

**Source data 1.** Raw unedited images of Co-immunoprecipitation analysis of ADNP with ZMYM2 (*Figure 1C*).

**Figure supplement 1.** ADNP interactions with ZMYM2.

*Figure 1 continued on next page*

*Figure 1 continued*

**Figure supplement 1—source data 1.** Raw unedited images of Co-immunoprecipitation analysis of ADNP with ZMYM3 (*Figure 1—figure supplement 1B*).

**Figure supplement 1—source data 2.** Raw unedited images of Western blot of ADNP expression in U2OS cells following treatment with siADNP or a non-targeting (NT) siRNA (top) (*Figure 1—figure supplement 1C*).

whose homologue BEND2 binds to ZMYM2 in mouse testes (*Ma et al., 2022*). However, 10 ZMYM2 binding proteins were uniquely identified here, including SUMO2 and SUMO3. The latter discovery is consistent with findings from screens for SUMO binding proteins where ZMYM2 was identified (*Aguilar-Martinez et al., 2015*; *Brüninghoff et al., 2020*). We validated interactions between endogenous ZMYM2 and ADNP by co-immunoprecipitation analysis (*Figure 1C*) and also confirmed interactions of ADNP with the closely related ZMYM3 paralogue (*Figure 1—figure supplement 1B*). This data is consistent with the identification of ZMYM2 and ZMYM3 in other mass spectrometry datasets along with ADNP (*Ostapcuk et al., 2018*; *Connaughton et al., 2020*). We also detected the ADNP paralogue ADNP2 as a ZMYM2 interactor.

Having established a physical interaction between ZMYM2 and ADNP, we next studied their functional interplay. First, we performed two independent ChIP-seq experiments for ADNP in U2OS cells and uncovered 5108 peaks found in both datasets. These ADNP peaks showed enrichment of binding motifs for several transcription factors with the top two motifs for HBP1 and IRF both found in over 35% of target regions (*Figure 1D*). The top scoring motif for ADNP previously identified from ChIP-seq in mouse embryonic stem cells CGCCCYCTNSTG (*Ostapcuk et al., 2018*), was not identified in this unbiased analysis. We therefore searched for the motif in the ADNP peaks and found 775 in 5108 peaks (15%) which was a substantially higher frequency than in a background dataset (9.5%) but well below the 63% frequency found in mouse embryonic stem cells. This motif was therefore present but at a relative low frequency and lowly enriched, suggesting an alternative chromatin recruitment mechanism. Co-binding on chromatin with ZMYM2 is inferred as there is a significant overlap in chromatin occupancy of ZMYM2 with the ADNP peaks (*Figure 1E*). Example loci show clear co-binding of the two proteins (*Figure 1F*).

Given this co-occupancy on chromatin we asked whether ZMYM2 and ADNP shared gene regulatory activity. We depleted each protein individually (*Figure 1—figure supplement 1C* and *Figure 4—figure supplement 1G*) and performed RNA-seq to monitor transcriptional changes. Depletion of either factor alone gave roughly equivalent numbers of up and downregulated genes (*Supplementary file 4*). However, there was a highly significant overlap in the genes upregulated following depletion of either factor, and also a highly significant overlap in the genes downregulated following reductions in either protein (*Figure 1G*). Conversely, in contrast to these similarities, gene regulation showing reciprocal directionality following either ADNP or ZMYM2 depletion showed a very small and insignificant overlap (*Figure 1—figure supplement 1C*) and this is further emphasised by a scatter plot depicting this data (*Figure 1—figure supplement 1E*). Thus, there is a strong overlap in gene regulatory activity for ZMYM2 and ADNP, although the large numbers of genes directionally co-regulated by these two proteins (ie either positively or negatively) indicates no clear common role as either an activator or repressor.

Together these results demonstrate that ZMYM2 can function through a transcriptional regulatory complex containing ADNP to co-ordinately control gene activity.

## ZMYM2 binds to molecularly distinct chromatin regions

Next, we turned to the potential interplay between ZMYM2 and TRIM28. Previous studies demonstrated that many zinc finger transcription factors can recruit TRIM28 to chromatin to repress transcription (Reviewed in *Bruno et al., 2019*). However this activity has previously been attributed to TRIM28 binding to the KRAB repression domain associated with many zinc finger proteins, but ZMYM2 lacks such a domain. To explore whether TRIM28 could be found in the same regions of chromatin as ZMYM2 we compared ChIP-seq data for TRIM28 in U2OS cells with our ZMYM2 binding profile. We also included ChIP-seq for SUMO2/3 and a SUMO binding defective version of ZMYM2, ZMYM2-mutSIM2 (as we previously showed that SUMO is needed for recruitment of ZMYM2 to chromatin; *Aguilar-Martinez et al., 2015*), markers of active chromatin including ChIP-seq data for H3K18ac (*Chen et al., 2016*) and ATAC-seq data. We took a ZMYM2-centric view of the data and plotted the

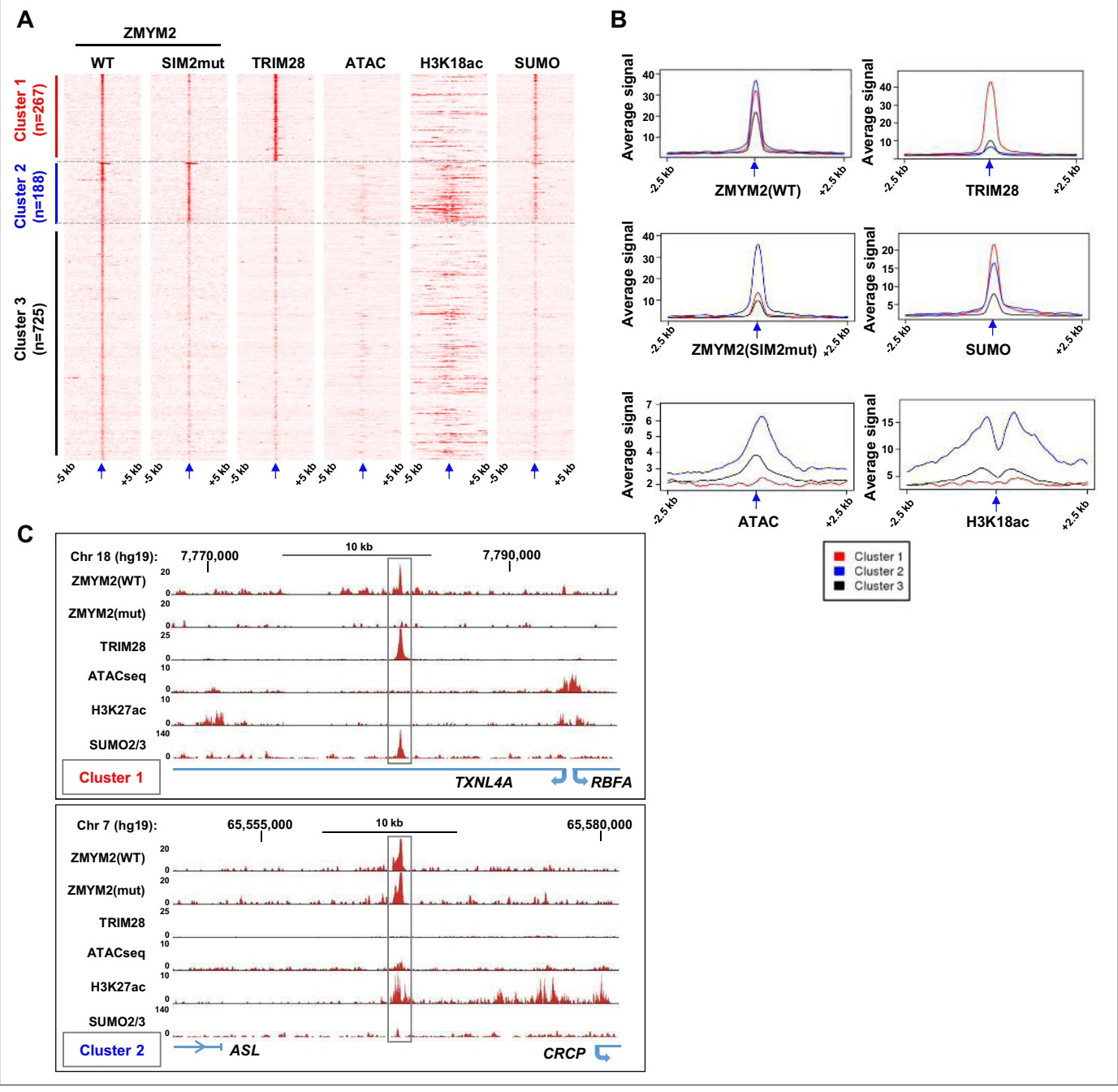

**Figure 2.** ZMYM2 interactions with molecularly distinct chromatin regions. (**A**) Heatmaps showing the signals of the indicated proteins or chromatin marks from ChIP-seq experiments or the ATAC-seq signal in U2OS cells plotted across a 10 kb region surrounding the centres (arrowed) of the wild-type (WT) ZMYM2 binding regions. Clustering of the data produced 3 clusters. (**B**) Tag density plots of the indicated ChIP-seq or ATAC-seq signals in the three clusters of ZMYM2 binding regions. (**C**) UCSC genome browser of the indicated ChIP-seq and ATAC-seq data on example genomic loci from clusters 1 and 2. See also *Figure 2—figure supplement 1*.

The online version of this article includes the following figure supplement(s) for figure 2:

**Figure supplement 1.** Molecular characterisation of ZMYM2 binding regions.

signal for the other chromatin binding/modification events centred on the ZMYM2 binding peaks and then clustered the data (*Figure 2A and B*). Three distinct clusters were revealed. Cluster 3 exhibits ZMYM2 binding but little evidence of any other binding/modification events. However, cluster 1 is enriched for both wild-type ZMYM2 and TRIM28 binding. This cluster is relatively depleted of

ATAC-seq and H3K18ac signal, suggesting that regions occupied by both ZMYM2 and TRIM28 are not areas of active chromatin (*Figure 2A and B*). Indeed, when we mapped H3K9me3 signal for H1 ESCs to the ZMYM2 binding regions, stronger levels of this repressive mark were observed in cluster 1 despite the differing cell types (*Figure 2—figure supplement 1A*). We validated H3K9me3 occupancy at ZMYM2 binding sites from cluster 1 in U2OS cells (*Figure 2—figure supplement 1B*). SUMO2/3 is also strongly detected in cluster 1 and SUMO has often been associated with transcriptional repressive activity (reviewed in *Garcia-Dominguez and Reyes, 2009*). We also interrogated ChIP-seq data from mouse ESCs and found that a cluster of ZMYM2 binding regions (cluster B) was strongly associated with TRIM28, SUMO and H3K9me3, further validating our results from human U2OS cells (*Figure 2—figure supplement 1C* and D). In contrast, ZMYM2 cluster 2 shows little TRIM28 binding and higher levels of ATAC-seq and H3K18ac signal (*Figure 2A and B*; *Figure 2—figure supplement 1E*), suggestive of more active chromatin. Conversely, binding of the SUMO binding defective ZMYM2(SIM2mut) is strong and SUMO occupancy is weaker suggesting that ZMYM2 activity at this cluster is independent of its SUMO binding activity. Example loci demonstrating the unique chromatin features associated with ZMYM2 binding regions in cluster 1 and cluster 2 are illustrated in *Figure 2C*.

Next we searched for over-represented binding motifs in each ZMYM2 cluster and found numerous significantly enriched motifs which differed between clusters 1 and 2 (*Figure 3A*). Interestingly, several of the motifs in cluster 2 are the same as in the peaks bound by ADNP (*Figure 1D*), including motifs for HBP1 and CTCFL/BORIS. The HBP1 motif was also previously observed as a preferred binding motif for the paralogue ZMYM3 (*Partridge et al., 2020*). Given the similarity in binding motifs, we superimposed the ADNP ChIP-seq signal onto the ZMYM2 clusters and found highest ADNP occupancy in cluster 2 as predicted from the motif analysis (*Figure 3B and C*). Cluster 3 ZMYM2 peaks showed enrichment of a different set of DNA binding motifs with ASCL2 and ETS transcription factor binding motifs figuring prominently (*Figure 3—figure supplement 1A*).

Differences were also observed in genomic location distributions of each cluster with cluster 1 peaks showing a predominantly intergenic or intronic distribution, whereas cluster 2 and to a lesser extent, cluster 3 peaks were more commonly located in promoter-proximal regions (*Figure 3D*; *Figure 3—figure supplement 1B*).

Together these results reveal two classes of ZMYM2 binding region with distinct molecular features. One is SUMO-dependent and is associated with inactive chromatin and TRIM28 co-binding. The second is SUMO-independent and is associated with more active chromatin and ADNP binding. Each type of region is characterised by enrichment of distinct DNA motifs, as well as different genomic features.

## ZMYM2 physically and functionally interacts with TRIM28

Next, we wanted to further understand the potential physical and functional interactions between ZMYM2 and TRIM28 implied from both RIME and chromatin co-occupancy in cluster 1 regions. First, we attempted to co-precipitate ZMYM2 and TRIM28 but were unable to detect co-binding under standard conditions (*Figure 4A*, lanes 4–6; *Figure 4—figure supplement 1A*). However, cluster 1 is also enriched for SUMO occupancy but depleted for binding of ZMYM2 containing a mutation in a critical SUMO interacting motif (SIM). The latter mutant was previously used to demonstrate the requirement for SUMO binding for ZMYM2 recruitment to chromatin (*Aguilar-Martinez et al., 2015*). We therefore repeated the co-precipitation experiment in the presence of NEM (an isopeptidase inhibitor and critical to preserve protein SUMOylation) and observed ZMYM2 binding to TRIM28 (*Figure 4A*, lanes 1–3; *Figure 4—figure supplement 1A*). Indeed, NEM addition stabilised high molecular weight SUMO conjugates and revealed the binding of ZMYM2 to SUMOylated protein(s) (*Figure 4—figure supplement 1B*). Further evidence for SUMO-dependent interactions was derived from co-IP studies where the SUMO binding defective ZMYM2(SIM2mut) showed weaker interactions with TRIM28 than wild-type ZMYM2 (*Figure 4—figure supplement 1C*). Interactions between ZMYM2 and TRIM28 in a cellular context were further validated using PLA (*Figure 4B*). These findings are consistent with the SUMO-dependent enrichment of TRIM28 occupancy on chromatin in ZMYM2 cluster 1 (*Figure 2A*) and this shared binding profile was validated by ChIP-qPCR (*Figure 4—figure supplement 1D*). This co-binding suggests that ZMYM2 might be responsible for recruiting TRIM28 to chromatin but we found no evidence for this as ZMYM2 depletion did not affect TRIM28 binding to a panel of its binding regions (*Figure 4—figure supplement 1E*). We also further validated our RIME experiments and

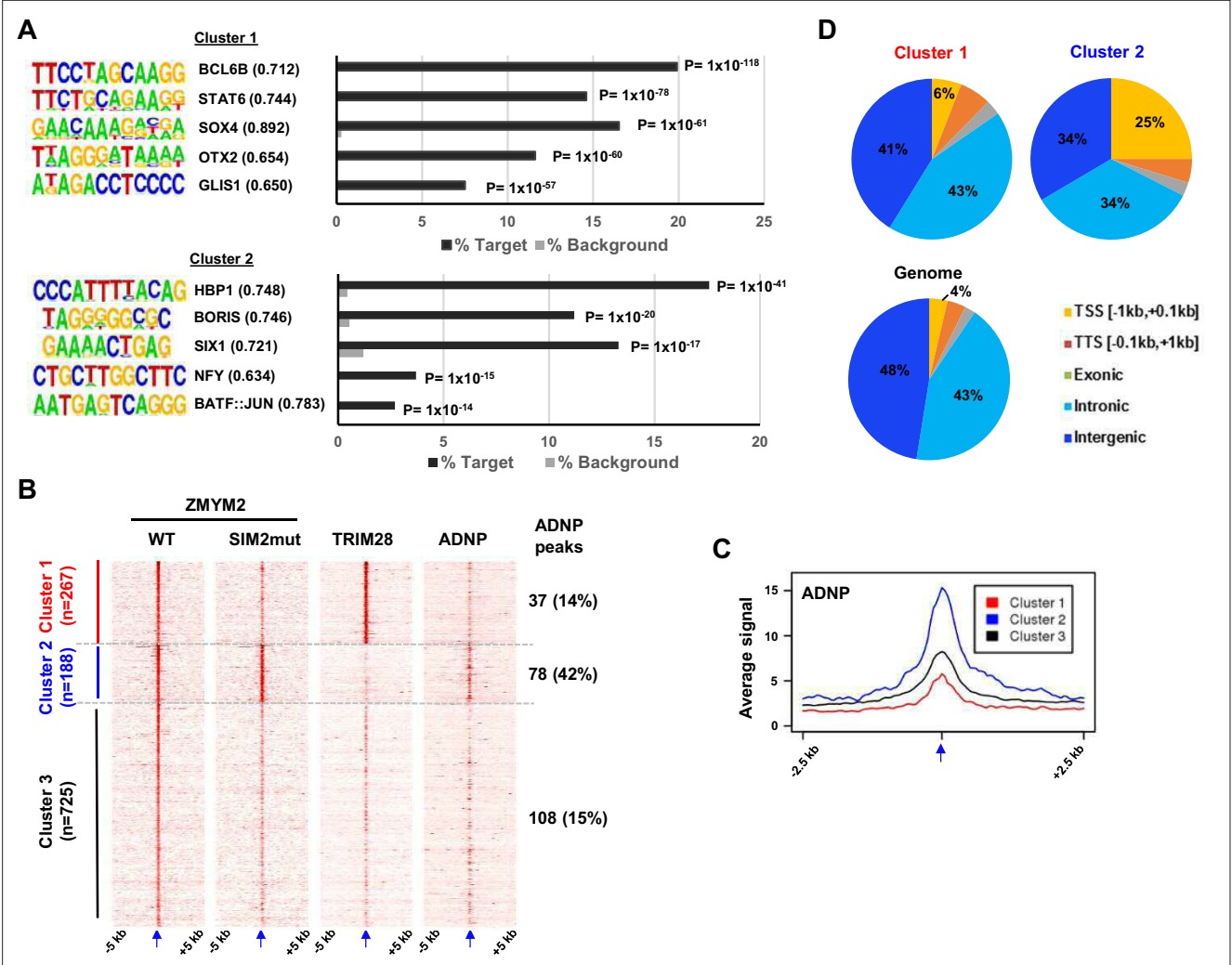

**Figure 3.** Characterisation of ZMYM2 chromatin binding regions. (**A**) De novo motif analysis of cluster 1 and 2 regions. The top five most significantly enriched motifs are shown with motif similarity to the indicated protein shown in brackets. (**B**) Heatmaps showing the signals of the indicated proteins from ChIP-seq experiments in U2OS cells plotted across a 10 kb region surrounding the centres (arrowed) of the wild-type (WT) ZMYM2 binding regions. Clustering was retained from *Figure 2A* and ADNP signal superimposed on top of this. The number and percentage of ZMYM2 peaks in each cluster overlapping with ADNP2 peaks is shown on the right. (**C**) Tag density plot of ADNP ChIP-seq signal from U2OS cells across a 5 kb region surrounding the centres (arrowed) of the three clusters of ZMYM2 binding regions. (**D**) Distribution of binding regions among different genomic categories for clusters 1 and 2 and the entire genome. See also *Figure 3—figure supplement 1*.

The online version of this article includes the following source data and figure supplement(s) for figure 3:

**Figure supplement 1.** Genomic distributions and motif enrichments in ZMYM2 binding regions.

**Figure supplement 1—source data 1.** Raw unedited images of Co-immunoprecipitation analysis of TRIM28 with ZMYM2 (*Figure 3—figure supplement 1A*).

**Figure supplement 1—source data 2.** Raw unedited images of Co-immunoprecipitation analysis of TRIM28 or SUMO2 with ZMYM2 (*Figure 3—figure supplement 1B*).

**Figure supplement 1—source data 3.** Raw unedited images of co-immunoprecipitation (IP) analysis of endogenous TRIM28 with the indicated EGFP-tagged ZMYM2 proteins (*Figure 3—figure supplement 1C*, top).

**Figure supplement 1—source data 4.** Raw unedited images of co-immunoprecipitation (IP) analysis of endogenous TRIM28 with the indicated EGFP-tagged ZMYM2 proteins (*Figure 3—figure supplement 1C*, bottom).

**Figure supplement 1—source data 5.** Raw unedited images of co-immunoprecipitation analysis of TRIM28 with ZMYM3 (*Figure 3—figure supplement 1F*).

**Figure supplement 1—source data 6.** Raw unedited images of western blot analysis of lysates from U2OS cells treated with the indicated targeting or non-targeting (NT) siRNAs.

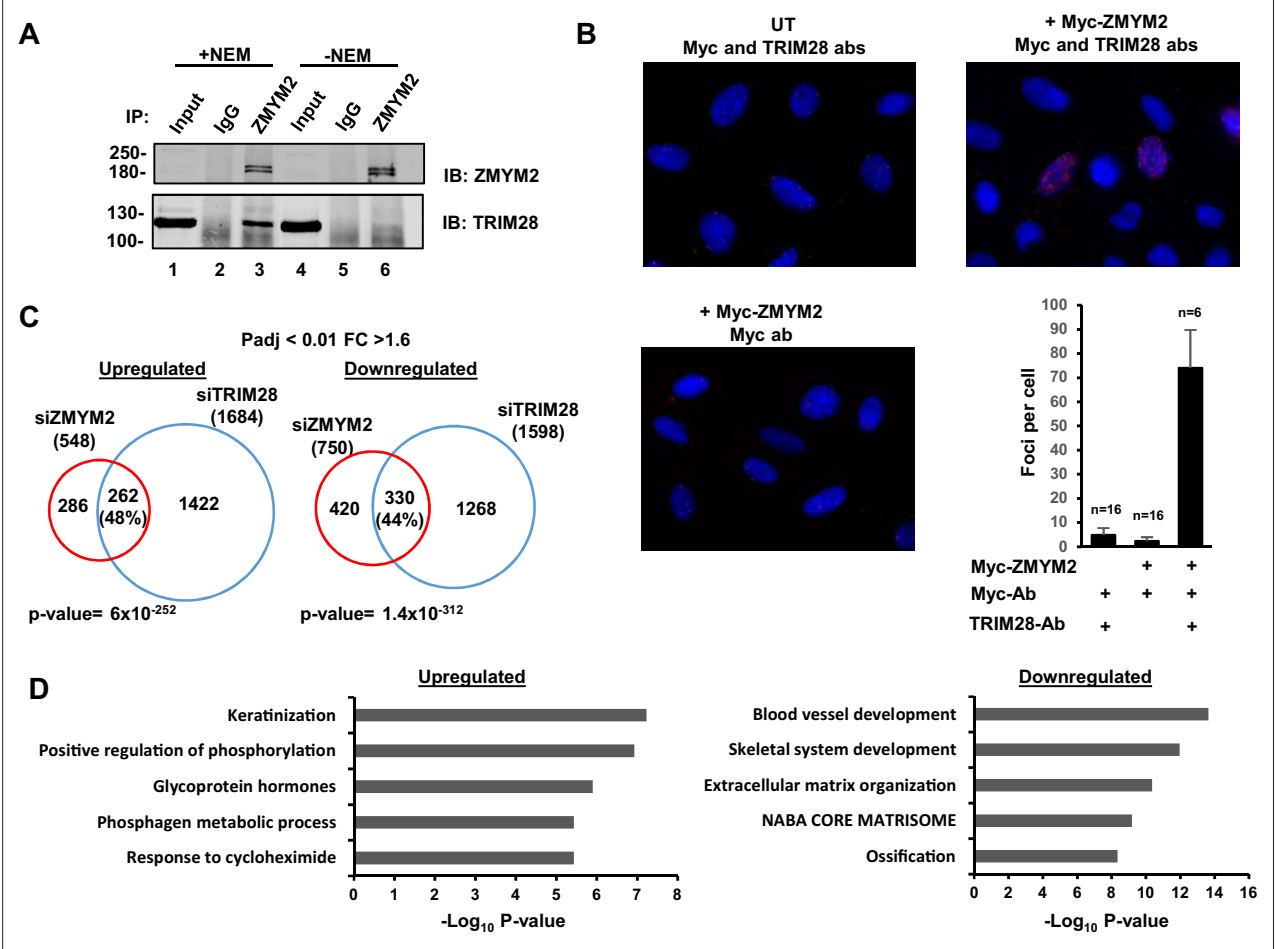

**Figure 4.** ZMYM2 interactions with TRIM28. (**A**) Co-immunoprecipitation analysis of TRIM28 with ZMYM2. Immunoprecipitation (IP) was performed with ZMYM2 or control IgG antibody from U2OS cells and the resulting proteins detected by immunoblotting (IB) with the indicated antibodies. 10% input is shown (See *Figure 4—figure supplement 1* for longer exposure). NEM was added to the extracts where indicated. (**B**) PLA assay of interactions between Myc-tagged ZMYM2 and endogenous TRIM28. Assays were carried out in U2OS cells transfected with a vector encoding Myc-ZMYM2 or left untransfected (UT). The addition of anti-Myc and -TRIM28 antibodies (ab) is indicated. Nuclei were stained with DAPI. Average numbers of foci per cell and numbers of cells are indicated (bottom). (**C**) Venn diagram showing the overlaps in upregulated (left) and downregulated (right) genes (Padj <0.05; fold change ≥1.6) from RNAseq analysis in U2OS cells treated with siRNAs against *ZMYM2* or *TRIM28*. (**D**) Enriched GO terms of genes commonly upregulated and downregulated by ZMYM2 and TRIM28 depletion. See also *Figure 4—figure supplement 1*.

The online version of this article includes the following source data and figure supplement(s) for figure 4:

**Source data 1.** Raw unedited images of Co-immunoprecipitation analysis of TRIM28 with ZMYM2 (*Figure 4A*).

**Figure supplement 1.** ZMYM2-TRIM28 interactions.

examined whether interactions with the ZMYM2 paralog ZMYM3 could be verified. Co-immunoprecipitation detected co-binding of TRIM28 with the ZMYM2 related protein ZMYM3 (*Figure 4—figure supplement 1F*) indicating that these interactions extend to other ZMYM2 paralogs.

The physical interactions between ZMYM2 and TRIM28, and their genomic co-occupancy, suggested that they might functionally interact. We therefore depleted each one in turn (*Figure 4—figure supplement 1G*) and performed RNAseq analysis to identify their target gene repertoires. For TRIM28, roughly equal numbers of genes were up and downregulated, and a similar phenomenon was observed for ZMYM2 following depletion, albeit more skewed to downregulation in the case of ZMYM2 (*Figure 4C*; *Supplementary file 4*). However, there was a substantial and highly significant overlap between ZMYM2 and TRIM28 regulated genes irrespective of the directionality (*Figure 4C*). Conversely, in contrast to these similarities, gene regulation showing reciprocal directionality following either TRIM28 or ZMYM2 depletion showed a very small and insignificant overlap

(*Figure 4—figure supplement 1H*). This is further emphasised by a scatter plot depicting this data (*Figure 4—figure supplement 1I*). Thus, functionally there is a strong concordance between the gene regulatory events controlled via ZMYM2 and TRIM28, although the large numbers of genes directionally co-regulated by these two proteins (ie either positively or negatively) indicates no particular preference for activation or repressive activity. Gene ontology (GO) analysis revealed different pathways associated with genes up and downregulated by both ZMYM2 and TRIM28 (*Figure 4D*). Terms such as 'ossification' and 'skeletal system development' were associated with the downregulated genes suggesting a disassembly of the core transcriptome associated with the cell identity (U2OS cells are derived from osteosarcomas), whereas upregulated genes contributed to more general GO terms, mainly associated with metabolism and cell responses such as 'positive regulation of phosphorylation'.

Together these results demonstrate that ZMYM2 and TRIM28 interact physically and functionally to control gene expression. However, these gene regulatory properties create a conundrum, as there is no obvious directionality in their effects, despite the fact that the proteins co-occupy regions of the chromatin with inactive/repressive features.

## ZMYM2 regulates protein coding genes from a distance

To further explore how ZMYM2 and TRIM28 work together to control gene expression, we tried to associate ZMYM2 cluster 1 peaks with genes commonly up- or down-regulated following ZMYM2 and TRIM28 depletion by using the nearest gene model. However, very few of these peaks are associated with the co-regulated genes identified from RNA-seq analysis (*Figure 5—figure supplement 1A*). This suggested that ZMYM2 might orchestrate gene activity from a distance. We further investigated this possibility by comparing the number of genes deregulated following ZMYM2 depletion that are associated with ZMYM2 peaks across the entire ZMYM2 binding dataset. Very few differentially expressed genes are found within 20 kb of a ZMYM2 peak (*Figure 5A*), but there is an increasing association as the window size is increased. The distance-dependent distribution of ZMYM2 binding regions is virtually the same as a randomly selected set of genes when considering genes that are downregulated following ZMYM2 depletion (*Figure 5A*, right). However, in contrast, significantly more differentially upregulated genes are associated with ZMYM2 peaks across all of the peak-gene distance brackets compared to a control set of genes (*Figure 5A*, left). This suggests a more direct role for ZMYM2 in transcriptional repression, albeit at a distance to target genes. Indeed when we focussed on ZMYM2 cluster 1 peaks (ie potentially ZMYM2-TRIM28 co-regulated) and the genes whose expression increased following either TRIM28 or ZMYM2 depletion (ie repressed by these factors), we found a more significant distance-based association of ZMYM2 peaks with upregulated rather than downregulated genes (*Figure 5B*). This is consistent with the observation that cluster 1 is predicted to be repressive in nature. To further explore this association, we examined the expression of all ZMYM2/TRIM28 co-regulated genes located within the TADs containing ZMYM2 binding peaks according to the three molecularly distinct clusters we identified. For this purpose, we used TADs from human ESCs (*Dixon et al., 2012*) on the well-established assumption that the majority of these will be conserved across cell types (*Stadhouders et al., 2018*). TADs containing cluster 1 binding regions contained a significantly higher number of upregulated genes compared to downregulated genes compared to TADs which lacked ZMYM2 binding regions (*Figure 5C*). Although there was a weak tendency for more upregulated genes with TADs containing cluster 2 peaks, this was insignificant. A similar trend was observed if genes were analysed based on their response to individual depletion of ZMYM2 or TRIM28, with TADs containing cluster 1 peaks being more associated with upregulated genes in both cases (*Figure 5—figure supplement 1B*). Moreover, we also tested whether a gene is more likely to be negatively regulated by ZMYM2 when located in the same TAD as a cluster 1 peak rather than being located in other TADs lacking such peaks. The observed frequency of co-localisation was significantly higher than by chance (*Figure 5—figure supplement 1C*). Although there is an association of ZMYM2 peaks with ZMYM2 regulated genes in the same TAD, we asked whether ZMYM2 might affect gene expression at a certain distance irrespective of whether the TSS of the gene is in the same TAD. To test this we took all of the ZMYM2 regions associated with genes upregulated by ZMYM2 depletion that resided in the same TAD and calculated the peak to TSS distance. Then we searched in the opposite direction for the TSS of genes at a similar distance (+/-25%) that resided in an adjacent TAD. We then asked whether these genes were upregulated by ZMYM2 depletion. 102 ZMYM2 peaks were positioned within these distance constraints with at least one gene in an adjacent

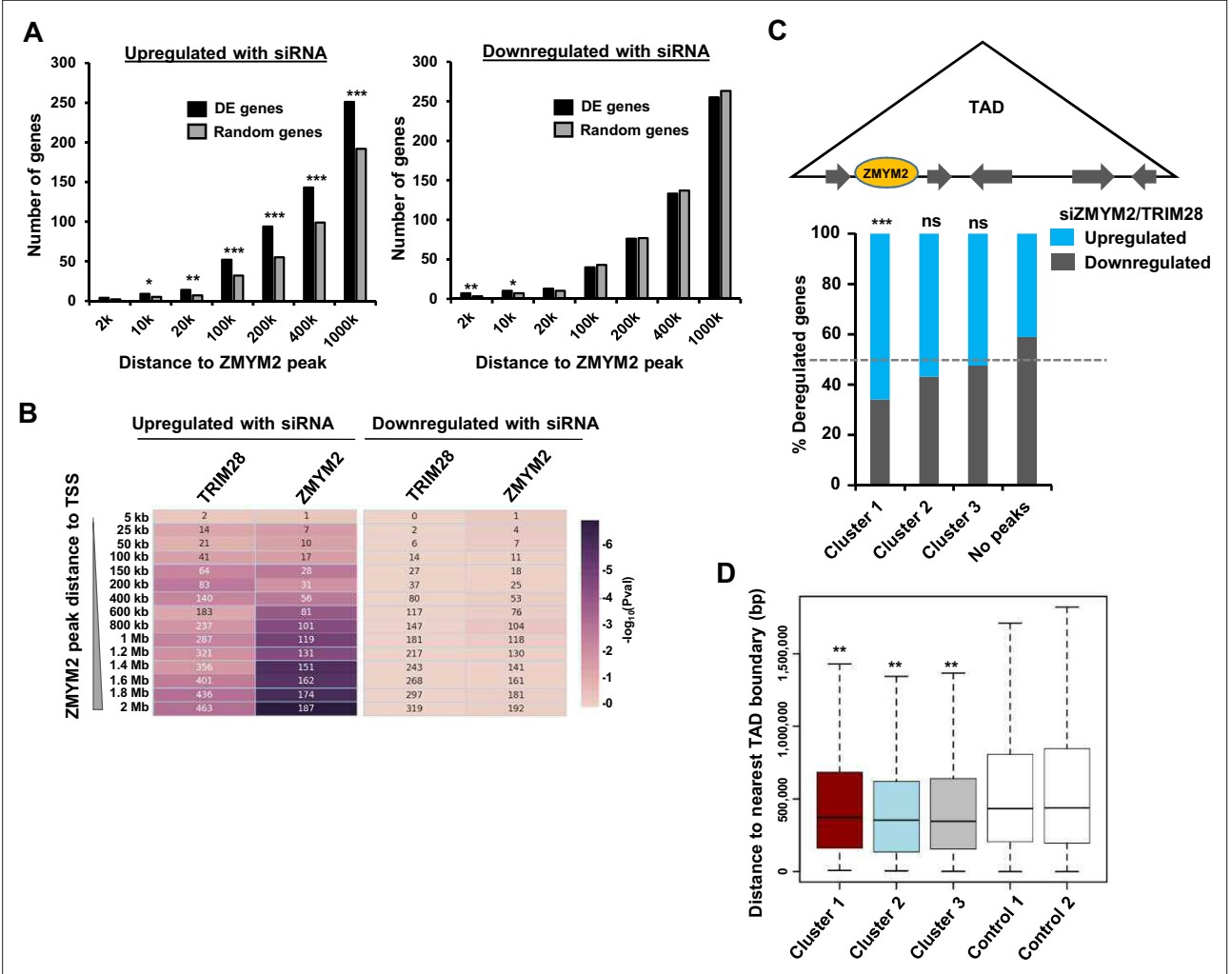

**Figure 5.** ZMYM2 location and gene expression. (**A**) Numbers of differentially expressed genes following ZMYM2 depletion (left upregulated, right downregulated) whose TSS lies within the indicated distances of a ZMYM2 binding peak (black bars). Control sets of equal numbers of randomly selected genes are shown for comparison (grey bars showing the average of 10 datasets). p-values *=<0.05, **=<0.01, ***=<0.001. (**B**) Distance-dependent association of ZMYM2 binding regions with differentially regulated genes following depletion of ZMYM2 or TRIM28. The numbers in the boxes are the number of genes among the input gene sets (x-axis) that overlap with the ZMYM2 peaks (cluster 1 peaks only) at the indicated distances to TSS (y-axis) and the colour shows $-\log_{10}$ of p-value (Hypergeometric test). (**C**) Relative proportion of genes commonly up- or down-regulated following ZMYM2 or TRIM28 depletion in TADs which also contain ZMYM2 peaks from the indicated clusters or have no ZMYM2 peak in the same TADs. Significance relative to regions containing no peaks is shown p-value, ***=<0.001; ns = non-significant. (**D**) Boxplots of the relative distance of ZMYM2 binding regions from each of the clusters from TAD boundaries compared to two different control sets of randomly selected regions (n=2360). Statistical significance of cluster 1–3 distances compared to each of the control regions is shown (**=p-value<0.05 in all cases; student t-test). See also *Figure 5— figure supplement 1*.

The online version of this article includes the following source data and figure supplement(s) for figure 5:

**Figure supplement 1.** ZMYM2 location and gene expression.

**Figure supplement 1—source data 1.** Raw unedited images of western blot illustrating the knockdown efficiencies of ZMYM2 in the RT-qPCR experiments.

TAD (716 genes in total). Of these genes, only 11 were upregulated following ZMYM2 depletion. There is therefore not a general spreading of transcriptional deregulation around ZMYM2 peaks in a distance-dependent manner. We further probed the association with TAD boundaries by comparing the distances of the ZMYM2 peaks in each of the clusters to TAD boundaries with randomly selected genomic regions. In all cases, the ZMYM2 peaks were significantly closer to a TAD boundary than expected by chance (*Figure 5D*).

Together, these results support a role for cluster 1 ZMYM2 peaks in gene repression, and suggest a more region-specific role rather than a simple peak-to-gene association typically found with transcription factors.

## ZMYM2 regulates retrotransposon elements

Several recent studies indicate a potential role for repetitive elements in controlling gene expression by demarcating TAD boundaries (reviewed in *Haws et al., 2022*) exemplified by the human endogenous retrovirus subfamily H (HERV-H)(*Zhang et al., 2019*). Zinc finger transcription factors, chiefly of the KRAB domain-containing subclass, have been associated with repetitive element repression (reviewed in *Bruno et al., 2019*). We therefore asked whether ZMYM2 binding regions mapped to any repetitive elements. All three clusters of ZMYM2 binding regions showed a high degree of overlap with repetitive elements, with 80% of the regions in cluster 1 showing close proximity to these elements (*Figure 6A*). However, the distribution of retrotransposon classes differed among ZMYM2 clusters, with cluster 3 and cluster 1 in particular favouring long terminal repeats (LTRs) whereas cluster 2 regions are mainly associated with short interspersed nuclear elements (SINE). The latter is consistent with the higher levels of ADNP binding found in cluster 2 regions, as ADNP has been implicated in SINE control (*Kaaij et al., 2019*). We further examined the distribution of subtypes of LTR elements in clusters 1 and 3 and found that cluster 3 is dominated by endogenous retrovirus (ERV)-L elements whereas cluster 1 is dominated by ERV-1 elements (*Figure 6B*). An example locus illustrates ZMYM2 and TRIM28 binding to a MER11A ERV-1 element (*Figure 6C*, top). Given the association of ZMYM2 with LTRs, we next asked whether ZMYM2 represses their activity as predicted. We depleted ZMYM2 and examined the regulation of LTR expression, and found that the LTRs showed upregulation following ZMYM2 loss, consistent with a repressive activity (*Figure 6D*; *Figure 6—figure supplement 1A*). We also examined the distribution of subclasses of SINEs associated with cluster 2 ZMYM2 peaks and found equal numbers of Alu repeats and MIR elements (*Figure 6E*). Importantly, depletion of either ZMYM2 (*Figure 6F*) or ADNP (*Figure 6—figure supplement 1B*) both caused increased activity of several SINEs, consistent with our designation of these retroviral elements as co-regulated by ZMYM2 and ADNP from our ChIP-seq analysis (eg *Figure 6C*, bottom). We also examined our RNA-seq data for more widespread activation of transposable elements beyond those we have tested, but were unable to detect expression of large numbers of additional elements. This may reflect a more limited effect of ZMYM2 on a subset of elements or that our data was of insufficient depth and quality to detect their expression.

Collectively, these results show that ZMYM2 associates with LTR containing regions, and promotes their repression in combination with the corepressor protein TRIM28 whereas it works alongside ADNP to control SINE element expression. Coupled with the observation that ZMYM2 binding is associated with regional control of gene expression and that retrotransposons can participate in indirect control of gene expression by affecting genome topology, this provides a plausible mechanism through which ZMYM2 impacts widely on coding gene expression.

## Discussion

ZMYM2 has been established as an important regulator of cell fate identity and commitment, particularly at the pluripotency stage. Genetic disruptions of ZMYM2 function have been linked to disease. Here, we provide further insights into the gene regulatory activities of ZMYM2 and demonstrate its physical and functional association with different corepressor complexes containing either ADNP or TRIM28 (*Figure 6G*).

We identify three distinct clusters of chromatin regions where ZMYM2 binds. One has chromatin characteristics related to transcriptional repression, one appears to be associated with more active chromatin regions and a third set of regions has no distinguishing chromatin features. It remains unclear what function, if any, ZMYM2 has at the latter set of binding regions. Our main focus was on cluster 1 ZMYM2 binding regions that are also bound by a novel binding partner, TRIM28. Interestingly, ZMYM2 binding to TRIM28 is SUMO-dependent, as retention of SUMOylation is required to detect binding by co-IP and ZMYM2 binding to cluster 1 regions require the SUMO interaction motifs in ZMYM2. This is consistent with the preponderance of SUMO at cluster 1 binding regions. It is not currently clear what SUMO modified protein(s) are the direct binding target of ZMYM2, although

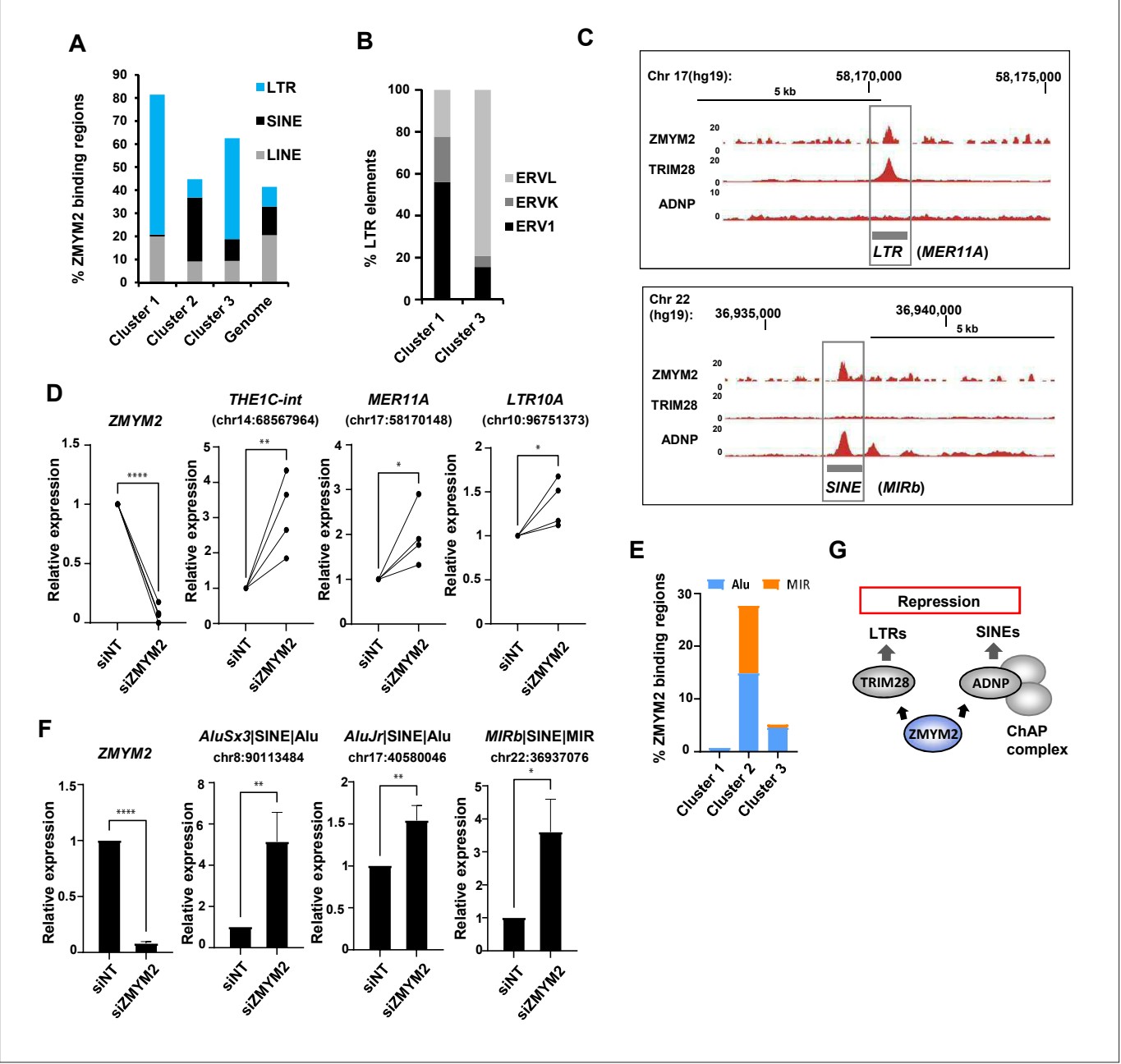

**Figure 6.** ZMYM2 functionally associates with ERV repetitive elements. (**A**) Percentage of the ZMYM2 binding sites in each of the indicated clusters containing each of the indicated classes of retrotransposon elements. The genome-wide proportion of genomic regions containing each type of the retrotransposon elements is also shown. (**B**) Proportions of LTR subclasses in cluster 1 and cluster 3 ZMYM2 binding regions. (**C**) UCSC genome browser view of a MER11A ERV1 LTR element located upstream of the *HEATR6* locus, illustrating co-binding of ZMYM2 and TRIM28 (top) and a MIR SINE element located upstream of the *EIF3D* locus, illustrating co-binding of ZMYM2 and ADNP. (**D**) RT-qPCR analysis of expression of ZMYM2 and the indicated LTR elements following ZMYM2 depletion or control non-targeting (NT) siRNA treatment. Individual paired experiments are shown (n=4; p-values *=<0.05,**=<0.01, ****=<0.0001). (**E**) Proportions of SINE subclasses in cluster 1–3 ZMYM2 binding regions. (**F**) RT-qPCR analysis of expression of ZMYM2 and the indicated SINE elements following ZMYM2 depletion or control non-targeting (NT) siRNA treatment (n=3; unpaired T-test p-values *=<0.05, **=<0.01, ****=<0.0001). (**G**) Model illustrating the two distinct complexes through which ZMYM2 functions on chromatin to control retrotransposon expression. See also *Figure 6—figure supplement 1*.

The online version of this article includes the following figure supplement(s) for figure 6:

**Figure supplement 1.** ZMYM2 containing complexes regulate retrotransposons.

ZMYM2 is itself SUMOylated (*Kunapuli et al., 2006*), as is TRIM28 (*Lee et al., 2007*), which can also act as a SUMO E3 ligase (*Ivanov et al., 2007*). Earlier studies may have missed the ZMYM2-TRIM28 interactions by performing ZMYM2 immunoprecipitations in the absence of SUMO stabilising agents or sample fixation as we used in our initial RIME experiments. However, ZMYM2 was identified in reciprocal co-IP experiments with TRIM28, as a glycosylated interaction partner (*Boulard et al., 2020*) suggesting that multiple post-translational modifications govern the assembly and function of this complex.

Previous studies have focussed on the role of ZMYM2 in the context of its association with the LSD1-CoREST-HDAC corepressor complex (*Hakimi et al., 2003*; *Shi et al., 2005*; *Gocke and Yu, 2008*). Indeed, we identify all three of these proteins in our proteomics dataset (*Supplementary file 3*), further substantiating this functional interaction. Here we find a broader occurrence of ZMYM2 in chromatin binding complexes, both as part of the ChAHP complex centred on ADNP (*Ostapcuk et al., 2018*) and also independently in a complex with TRIM28. The latter observation is interesting, as TRIM28 has been shown to act as a widespread repressor of retrotransposons where it is recruited by KRAB domain containing zinc finger proteins (reviewed in *Bruno et al., 2019*). ZMYM2 is also a zinc finger protein but lacks the KRAB domain but still binds to TRIM28, suggesting that a broader swathe of zinc finger proteins may be involved in repetitive element epigenetic repression. However, our results rule out a simple model that ZMYM2 is responsible for the recruitment of TRIM28 to chromatin. Whether KRAB-zinc finger proteins are also involved in ZMYM2 complexes and act as the relevant recruitment factor is currently unknown although we identified four such proteins in our lower confidence proteomics dataset, with ZNF202 being the most prominent consistently detected protein (*Supplementary file 3*). It is also not clear what the chromatin recruitment mechanism is, although recent studies do not rule out a direct role for TRIM28 in the process as it has been shown to be able to engage directly with unmodified histone H4 tails (*Bacon et al., 2020*).

The identification of TRIM28 as a chromatin-associated interaction partner further supports a role for ZMYM2 in transcriptional repression, and suggests that it can do so via a variety of different means. Indeed a recent study identified ZMYM2 as an important regulator of silencing of imprinting control regions which are under the control of the zinc finger protein ZFP57 (*Butz et al., 2022*). In terms of global transcriptional regulation, the role of ZMYM2 is more ambivalent as we identified substantial numbers of genes that are either up- or down-regulated. However, ZMYM2 binding regions are generally more associated with genes that it represses rather than activates, suggesting that the latter may be indirectly affected. In contrast to TRIM28 co-bound regions, ZMYM2 also binds to chromatin regions occupied by ADNP, where more active characteristics are found. ZMYM2 functions in a repressive manner rather than in gene activation in the latter context and suggests that it dampens down the activity of a region rather than the complete repression which would be mediated through TRIM28 and its coregulatory partners.

The association of ZMYM2 with transposons and its ability to regulate their expression suggests several possible modes of subsequent effects on broader gene expression programmes. Retrotransposons have been shown to exhibit a multitude of different molecular mechanisms to control gene regulation (reviewed in *Fueyo et al., 2022*). For example, the ERV-H LTRs demarcate TADs in human pluripotent stem cells (*Zhang et al., 2019*), and it is possible that ZMYM2-TRIM28 complexes control ERV LTR activity which in turn influence local chromatin topology. This would help explain how ZMYM2 appears to regulate gene expression in a locally defined region rather than one ZMYM2 binding event acting to regulate expression of one specific gene. This might apply in particular to the ZMYM2-ADNP co-bound regions, as ADNP has previously been shown to play an active role in controlling chromatin domain boundary elements at SINE B2 transposable elements by blocking CTCF binding (*Kaaij et al., 2019*). It is also noteworthy that ZMYM2 interacts with the GTF3C complex which has previously been implicated in controlling retrotransposon transcription (*Ferrari et al., 2020*). Thus, ZMYM2 may contribute important regulatory activities to many different chromatin complexes associated with transposable elements. However, further extensive work will be needed to address the precise interplay between ZMYM2, retrotransposon activity, and local chromatin structure in determining ZMYM2-dependent gene regulatory outcomes.

In summary, our work adds to the growing repertoire of ZMYM2 associated chromatin regulatory proteins and complexes and provides a molecular framework for understanding the role of this protein in cell fate commitment and maintenance.

# Materials and methods

## Cell culture

U2OS and HEK293T cells and derivatives were grown in DMEM supplemented with 10% fetal bovine serum. Cells were cultured at 37 °C, 5% $CO_2$ in a humidified incubator and were passaged every 3–4 days using 0.05% Trypsin-EDTA (ThermoFisher Scientific, 25300). Stable U2OS-derived cell lines, flag-ZMYM2(WT) and SUMO3(K11R/Q90P) were created using U2OS-Flp-in cells (*Aguilar-Martinez et al., 2015*). U2OS and HEK293T cells were originally obtained from ATCC and were routinely checked for mycoplasma contamination.

## RIME Mass Spectrometry analysis

U2OS-derived cells were grown in 15 cm dishes (6 confluent dishes per sample) and fixed using DMA for 20 min at room temperature, followed by 1% formaldehyde (ultra pure) addition for 5 min at room temperature. Cells were washed twice with cold 1 X PBS. Cells were harvested in 1 X PBS supplemented with protease inhibitors (cOmplete cocktail, 11873580001, Roche). Cells were centrifuged at 3000 rpm for 10 min at 4 °C. The supernatant was removed, the cell pellet was resuspended in 1 X PBS with protease inhibitors and spun down again. The pellets were resuspended in LB1 buffer (50 mM Hepes-KOH pH 7.5, 140 mM NaCl, 1 mM EDTA, 10% glycerol, 0.5% Igepal, 0.25% Triton X-100 supplemented with cOmplete cocktail (11873580001, Roche), 20 mM N-ethylmaleimide (NEM) and 30 mM iodoacetate (IAA)), and incubated with gently agitation for 10 min at 4 °C. Cells were spun down at 2000 *g* for 5 min at 4 °C. Pellets were resuspended in 10 ml of LB2 buffer (10 mM Tris-HCl pH8, 200 mM NaCl, 1 mM EDTA, 0.5 mM EGTA, supplemented with cOmplete cocktail (11873580001, Roche), 20 mM NEM and 30 mM IAA) incubated with gently agitation for 5 min at 4 °C. Samples were spun down at 2000 *g* for 5 min at 4 °C. Pellets were resuspended in LB3 (10 mM Tris-HCl pH 8, 100 mM NaCl, 1 mM EDTA, 0.5 mM EGTA, 0.1% Na-Deoxycholate, 0.5% N-lauroylsarcosine, supplemented with cOmplete cocktail (11873580001, Roche), 20 mM NEM and 30 mM IAA), aliquoted and sonicated; 15 cycles of 30 s sonication, 30 s paused, in a water bath at 4 °C. After sonication, triton X-100 was added to the samples. Samples were spun down at 13,000 rpm for 10 min at 4 °C. The supernatant was removed and incubated with antibody coupled beads. Samples were incubated with gentle agitation over night at 4 °C. Beads were washed 10 times with 1 ml of RIPA buffer and twice with freshly made 100 mM ammonium bicarbonate. Beads were spun down at 1000 rpm for 1 min to remove residual ammonium bicarbonate. To release the proteins bound to the beads, 100 ng of trypsin was added and incubated overnight at 37 °C. The supernatant was carefully removed and analysed by an Orbitrap velos mass spectrometer as described previously (*Aguilar-Martinez et al., 2015*).

The data produced was searched using Mascot (Matrix Science, version 2018_01), against the SwissProt (version 2018_01) with a taxonomic break of the Human (*Homo sapiens*) database. Data was validated using Scaffold (5.0.2). Proteins found in more than 50% of CRAPome samples, ribosomal proteins and proteins with zero spectral counts in any experiment were removed from the list. Proteins showing an average of >4 spectral counts across three experiments were included in the final list and were then compared to proteins bound to ZMYM2 on mouse ESCs (*Yang et al., 2020*) and BioID experiment on 293 cells (*Connaughton et al., 2020*).

## RNA interference, RT-qPCR and RNA-seq analysis

ON-TARGETplus SMART pool siRNAs from Dharmacon were used to knock-down ZMYM2, TRIM28 or ADNP (L-021348-00-0005, L-005046-00-0005 and L-012857-01-0005 respectively). Non-targeting siRNA pool (siNT, Dharmacon, D-001810-10-20) was used as a control. To carry out RNA interference (RNAi), cells were transfected with siRNA using *Lipofectamine RNAiMAX* (Invitrogen) according to the reverse transfection method in the manufacturer's instructions. Forty-eight hr after transfection, cells were transfected again with the same siRNA using forward transfection method for another 48 hr.

Total RNA was extracted using the RNeasy plus kit (Qiagen) following the manufacturer's protocol. To detect LTR expression, additional DNase digestion was performed using RNase-Free DNase Set (Qiagen) to completely remove genomic DNA.

For RNA-seq analysis, U2OS cells in 6-well plates were transfected twice, second transfection was done 48 hr after the first one. A final concentration of 100 nM of each siRNA using Lipofectamine RNAiMAX (ThermoFisher Scientific), were used following the manufacturer's instructions. Total RNA was extracted 48 hr after the second transfection using an RNeasy plus kit with on-column DNase

treatment, following the manufacturer's instructions (Qiagen). Immuno-blot and RT-qPCR for ZMYM2, TRIM28 or ADNP expression were carried out to verify knock-down of the mRNAs and proteins.

Quantitative RT-PCR was performed using 30 ng of total RNA and QuantiTecT SYBR Green RT-PCR kit (Qiagen, 204243) according to the supplier's protocol. Data were analysed by Qiagen Rotor-Gene Series software. Data were normalised against the control gene *GAPDH* (primers ADS2184 ACAG TCAGCCGCATCTTCTT and ADS2185 TTGATTTTGGAGGGATCTCG) or *RPLP0* (ADS5454, ATTA CACCTTCCCACTTGCT and ADS5455 CAAAGAGACCAAATCCCATATCCT). RNA samples were run in duplicate from at least three independent experiments. The primer-pairs used for RT-qPCR experiments were *ZMYM2* (ADS3573, GGTAAACTAACTGAGATTCGCCA and ADS3574, CCAGAACATTAT TTCCAGCACCT), *TRIM28* (ADS6181, GTTTCAGTGGGACCTCAATGC and ADS6182, GATCATCT CCTGACCCAAAGC), *ADNP* (ADS6688, AGGCTGACAGTGTAGAGCAAG and ADS6689, GACT GCCCCATTGAGTGATTTT), *MER11A* (ADS6917, AATACACCCTGGTCTCCTGC and ADS6918, AACA GGACAAGGGCAAAAGC), *LTR10A* (ADS6919, ACAACTTTCCCACCAGTCCT and ADS6920, GCAG GAGTATGAGCCAGAGT), *THE1C-int* (ADS6543, CCAACCCGACATTTCCCTTC and ADS6544, AGGG GCCAAGGTACATTTCA), *AluSx3* (ADS6937, TGAGGTGGGCTGATCATGAG and ADS6938, TGCA ACCTCCACCTCCTAAG), *AluJr* (ADS6939, AGGCTGAGTTGGGAGGATTG and ADS6940, GCAG GATCTCATTCTGTTGCC), *MIRb* (ADS6941, AGTGCCAGCTTTGGTTTCAG and ADS6942, AGAC GAGAACACTGAGGCTC).

## Western blot analysis and co-immunoprecipitation assays

For immuno-blotting, the antibodies used are listed in *Supplementary file 1*. Lamin-B and Tubulin were used as loading controls. Proteins were visualised using IRDye infrared dye (Li-Cor Biosciences) conjugated secondary antibodies and the signals were visualised using a LI-COR Odyssey CLx scanner and analysed using ImageStudio v 5.2.5.

Co-immunoprecipitation was performed as previously described (*Ji et al., 2012*). The anti-ZMYM2, ZMYM3 and ADNP antibodies (*Supplementary file 1*) were used for immunoprecipitation, with normal rabbit IgG (Millipore) as negative control. For co-immunoprecipitation analysis of endogenous TRIM28 with the EGFP-tagged ZMYM2 proteins, HEK293T cells were transfected with EGFP vector, EGFP-ZMYM2(WT) (pAS4329) or EGFP-ZMYM2(SIMmut) (pAS4330) plasmids using Polyfect Transfection Reagent (Qiagen, Cat. No. ID: 301105) for 24 hr, cell lysates were prepared in the presence of cOmplete cocktail (11873580001, Roche) and 20 mM N-ethylmaleimide (NEM) and followed by immunoprecipitation using TRIM28 antibody.

## In situ *Proximity ligation assay (PLA)*

U2OS cells grown on glass cover-slips were washed once in 1 X PBS and then fixed and permeabilised with 3.7% para-formaldehyde plus 0.8% Triton X-100 for 15 min at room temperature. Cells were washed three times with 1 X PBS. During all of the incubation steps cells were kept in a humidified chamber. The cells were first incubated for one hour with 1% BSA dissolved in 1 X PBS, followed by one hour incubation with both primary antibodies (anti-c-myc Santa Cruz (9e10):sc-40 1:500, and anti-KAP1 Abcam, ab10483 1:500, diluted in 1% BSA-1X PBS). Cells were washed three times with 1 X PBS and once with PBS[+] (1 X PBS, 0.1% Tween-20, 1% BSA). Cells were incubated with PLA probes, ligase and polymerase following Duolink-II fluorescence instructions. After the final wash, nuclei were stained by incubating with Hoechst solution (1 μg/ml) for 10 minutes at room temperature. Cells were washed twice with 1 X PBS. Preparations were mounted in vectashield antifade medium (Vector) and sealed with nail varnish. Z-stack images covering the entire thickness of the samples were acquired on a Delta Vision RT (Applied Precision) restoration microscope using a 100 X/1.40 Plan Apo objective. The images were collected using a Coolsnap HQ (Photometrics) camera with a Z optical spacing of 0.2 μm. Raw images were then deconvolved using the Softworx software. Deconvolved images were used to quantify interactions per cell using the single cell analysis function of Blobfinder software.

## RNA-seq data analysis

RNA-seq was performed in triplicate. Libraries were generated using TruSeq stranded mRNA library kit (Illumina) and sequenced on HiSeq 4000 platform (Illumina). Reads were mapped to the human genome hg19 by the aligining tool STAR as described previously (*Dobin et al., 2013*; *Ji et al., 2021*). 14.6 M-43.4 M reads were uniquely mapped for the three replicates of siZMYM2, 18.7 M-48.5 M reads

were uniquely mapped for the three replicates of siTRIM28, and 22.6 M-23.1 M reads were uniquely mapped for the three replicates of siADNP (**Supplementary file 2**). Differentially regulated genes between two different conditions were established using the R package edgeR (**Robinson et al., 2010**) with the criteria adjusted *P*-value <0.01 and fold change >1.6. The online tool Metascape (**Zhou et al., 2019**) was used for GO terms and pathway enrichment analysis on the DE gene sets.

### ATAC-seq analysis

The ATAC-seq data in U2OS cells were generated by using the protocol essentially as described previously (**Yang et al., 2019**). ATAC-seq libraries were generated from U2OS cells treated with a non-targeting siRNA pool. Cells were lysed 48 hr after transfection in cold lysis buffer (10 mM Tris-HCl, pH 7.4, 10 mM NaCl, 3 mM MgCl$_2$ and 0.1% Igepal). Nuclei were concentrated by centrifugation at 500 *g* for 10 min at 4 °C. The nuclei pellet was resuspended in nuclease free water, diluted 1:10 and quantified. The equivalent volume for 50,000 nuclei was made up to 22.5 µl with nuclease free water. Nuclei were incubated for 30 min at 37 °C, 300 rpm, with 25 µl of 2 x TD buffer and 2.5 µl tagmentation enzyme (Illumina transposase FC-121–1030). DNA was purified using a Qiagen mini elute kit following the manufacturer's instructions. Library fragments were amplified using standard PCR and Nextera primers (Illumina) by adding 25 µl of 2 x NEBnext master mix, 2.5 µl forward primer, 2.5 µl reverse primer and 20 µl transposased DNA. PCR was run for 5 cycles, 72 °C, 5 min; 98 °C 30 s; cycle of 98 °C 10 s, 63 °C 30 s, 72 °C 1 min, paused at 4 °C. To determine the total number of cycles needed for the amplified library, 5 µl of the PCR mix were taken and added to a new mix containing 5 µl 2 X master mix, 1 ml of each primer, 0.6 µl of 10 X SYBR green and 2.4 µl H$_2$O. In a qPCR machine, 20 cycles were run, the total number of cycles for the ATAC library was chosen according to qPCR cycle that gave 1/3 of saturation. DNA was purified using Ampure XP beads following the manufacturer's instructions.

The reads in the ATAC-seq data were mapped to the human genome hg19 using Bowtie2 with the same settings as those described below for the ChIP-seq data, and only the uniquely mapped reads were used in the analysis.

## Chromatin immunoprecipitation (ChIP)-qPCR and ChIP-seq assays

ChIP-qPCR was performed using primers to detect *TXNL4*, *EMX2*, and *FAM109A*, as described previously (**Aguilar-Martinez et al., 2015**), *MSTB* (ADS6541, GACCAGCCTGACCAAAACG and ADS6542, ACTCAGGCCAAGTCTCTCTC), and *THE1C-int* (ADS6543, CCAACCCGACATTTCCCTTC and ADS6544, AGGGGCCAAGGTACATTTCA).

5x10$^7$ U2OS cells were used in each ChIP-seq experiment, which was performed essentially as described previously (**Ji et al., 2012**) with anti-TRIM28 antibody (**Supplementary file 1**) as indicated. There was minor modification in immuno-precipitation step for the ADNP ChIP-seq experiments, which was performed overnight at 4 °C by incubating the shared DNA-protein complex with 10 µg of anti-ADNP (Abcam Ab231950) antibody, followed by adding Dynabeads protein A (Invitrogen) and incubating for further 2 hr.

Immunoprecipitated DNA was purified with a PCR purification kit (Qiagen) and approximately 10 ng of DNA were processed for sequencing.

## ChIP-seq data processing and analysis

The data analysis of both ZMYM2(WT) and ZMYM2(SIM2mut) ChIP-seq data were described previously (**Aguilar-Martinez et al., 2015**), including mapping the reads to human genome hg19 and obtaining 1188 peaks from the ZMYM2(WT) ChIP-seq data. We removed 8 peaks because those 8 peaks have very high signal in all the ChIP-seq data we examined and one input DNA data of U2OS cells (also in **Aguilar-Martinez et al., 2015**), and used 1180 ZMYM2 peaks in all the analysis in the paper. The R method *kmeans* with the option 'centers = 3' was used to cluster the 1180 ZMYM2 peaks into three clusters.

For TRIM28, ADNP and SUMO ChIP-seq analysis, two biological replicates were generated for each protein and sequenced using the Illumina HiSeq4000 platform. The paired-end reads were aligned to the human genome hg19 using Bowtie2 (**Langmead and Salzberg, 2012**) with the setting 'very sensitive' and all other default settings. The uniquely aligned reads were selected for the further analysis. Duplicate reads were removed using Picard tools, version 1.1 (http://broadinstitute.github.io/picard/) and the reads mapped to the mitochondrial chromosome were also removed. The correlation

coefficients between the two replicates of the read counts was between 0.85 and 0.95 for the two replicates in our ChIP-seq data. The reads in the two replicates were then pooled, and with the pooled reads from the two input replicates as control, peaks were called from the pooled ChIP-seq using MACS2 (*Zhang et al., 2008*) with the options '-g hs -f BAMPE'. To identify overlapping peaks between two samples, one peak was considered to overlap with another peak if they overlap by at least 30% of each of their lengths. Venn diagrams showing the overlapping of peaks from different ChIP-seq datasets were created using the R package Vennerable, version 1.1 (https://github.com/js229/Vennerable; *js229, 2016*). Average tag density plots were generated as described previously (*Ji et al., 2021*).

The genomic distribution of the ChIP-seq peaks was calculated using all the transcripts in the Ensembl human gene annotation database v75 with the human genome hg19. The genome was divided into five types of regions; promoter or TSS regions [–1 kb to 0.1 kb relative to TSS], TTS regions [–0.1 kb to 1 kb relative to TTS], exonic regions containing all exons within [TSS +0.1 kb to TTS-0.1 kb], intronic regions containing all introns within [TSS +0.1 kb to TTS-0.1 kb], and all other regions were classified as 'distal intergenic' regions. A peak was linked to a region if its summit is located in the region.

The software HOMER (*Heinz et al., 2010*) was used with default settings for the motif enrichment analysis within a set of peaks (in 101 bp windows centred on the peak summits). The human genome annotations of LTR, SINE and LINE in hg19 contained in HOMER were also used in the analysis of their relationship to the ZMYM2 peaks.

To associate ZMYM2 or ADNP binding events with potential biological functions, we used the online tool GREAT (*McLean et al., 2010*) to obtain the GO terms for each set of ChIP-seq peaks.

## Statistical analysis

Statistical analysis for real-time PCR results was performed using the Student t test. The error bars in all graphs represent standard deviation. The statistical test for the ATAC-seq data signal (Fig. S2E) and the distance of the ZMYM2 peaks in different clusters to the nearest TAD domains were also performed using the Student t test. Fisher Exact test was used for testing the significance of the overlaps between different sets of DE genes. The software PEGS (*Briggs et al., 2021*) was used to assess the statistical significance of the association between the ZMYM2 peaks and the DE genes.

## Datasets

Our ChIP-seq and RNA-seq data from U2OS cells have been deposited with ArrayExpress. Accession numbers: E-MTAB-12292 (ADNP and TRIM28 ChIP-seq), E-MTAB-12293 (SUMO ChIP-seq), E-MTAB-12294 (ATAC-seq), and E-MTAB-12291(RNAseq following ZMYM2, TRIM28 or ADNP depletion).

The following existing datasets were used: H3K18ac ChIP-seq in U2OS cells (E-MTAB-3695), ZMYM2 ChIP-seq (WT and SIM2mut) in U2OS cells (E-MTAB-2701; *Aguilar-Martinez et al., 2015*), ZMYM2 ChIP-seq in mouse mESC CJ7 cells (GSM3384427, GSM3384428; *Yang et al., 2020*), Trim28 ChIP-seq in Mouse E14 ESCs (GSM3611260; *Seah et al., 2019*), H3K9me3 ChIP-seq in Mouse E14 ESCs (GSM3611253, GSM3611256, GSM3611258; *Seah et al., 2019*), SUMO ChIP-seq in mouse ESCs (GSM2629945, GSM2629946; *Cossec et al., 2018*), H3K27ac and H3K9me3 in human H1 ESCs (https://egg2.wustl.edu/roadmap/data/byFileType/alignments/consolidated/; *Kundaje et al., 2015*).

## Acknowledgements

We thank Karren Palmer, Mairi Challinor and Guanhua Yan, for excellent technical assistance; Ian Donaldson, Ping Wang, Rachel Scholey and Leo Zeef in the Bioinformatics core Facility; Stacey Holden, Michal Smiga and Andy Hayes in Genomic Technologies Core Facility; staff in the Mass spectrometry and Bioimaging facilities, Nicoletta Bobola, Shen-Hsi Yang and members of our laboratory for comments on the manuscript and stimulating discussions. This work was supported by the Wellcome Trust (103857/Z/14/Z) and BBSRC (BB/V000403/1).

## Additional information

### Funding

| Funder | Grant reference number | Author |
| --- | --- | --- |
| Wellcome Trust | 103857/Z/14/Z | Danielle J Owen |
| Biotechnology and Biological Sciences Research Council | BB/V000403/1 | Andrew D Sharrocks |

The funders had no role in study design, data collection and interpretation, or the decision to submit the work for publication. For the purpose of Open Access, the authors have applied a CC BY public copyright license to any Author Accepted Manuscript version arising from this submission.

### Author contributions

Danielle J Owen, Zongling Ji, Formal analysis, Investigation, Writing – review and editing; Elisa Aguilar-Martinez, Yaoyong Li, Formal analysis, Investigation, Writing – original draft, Writing – review and editing; Andrew D Sharrocks, Conceptualization, Supervision, Funding acquisition, Writing – original draft, Project administration

### Author ORCIDs

Zongling Ji  http://orcid.org/0009-0006-9133-615X
Andrew D Sharrocks  https://orcid.org/0000-0001-7395-9552

Reviewer #1 (Public Review): https://doi.org/10.7554/eLife.86669.3.sa1
Reviewer #2 (Public Review): https://doi.org/10.7554/eLife.86669.3.sa2
Reviewer #3 (Public Review): https://doi.org/10.7554/eLife.86669.3.sa3
Author Response https://doi.org/10.7554/eLife.86669.3.sa4

## Additional files

### Supplementary files

- Supplementary file 1. List of antibodies.
- Supplementary file 2. Sequencing statistics.
- Supplementary file 3. RIME analysis of the ZMYM2 interactome.
- Supplementary file 4. RNAseq analysis of gene expression changes following ADNP, ZMYM2 and TRIM28 depletion.
- MDAR checklist
- Source data 1. Original western blots used to create cropped figures shown in thje manuscipt.

### Data availability

UCSC browser session containing the peak tracks: http://genome.ucsc.edu/cgi-bin/hgTracks?db=hg19&position=chr1:18,078,462-18,084,961&hide=all&hgct_customText=http://bartzabel.ls.manchester.ac.uk/sharrockslab/yaoyong/ZNF198/index_file_hg19_chipSeq_ZMYM2_final.txt. Original ChIP-seq and ATAC-seq data from U2OS cells can be viewed on ArrayExpress at: E-MTAB-12292 (ADNP and TRIM28 ChIP-seq), E-MTAB-12293 (SUMO ChIP-seq) and E-MTAB-12294 (ATAC-seq).

The following datasets were generated:

| Author(s) | Year | Dataset title | Dataset URL | Database and Identifier |
|---|---|---|---|---|
| Aguilar-Martinez E, Owen D, Sharrocks AD | 2023 | RNA-seq of the U2OS cell line (Human Bone Osteosarcoma Epithelial Cells) treated with ZMYM2, TRIM28 or ADNP knock down against the control conditions | https://www.ebi.ac.uk/biostudies/arrayexpress/studies/E-MTAB-12291 | ArrayExpress, E-MTAB-12291 |
| Ji Z, Owen D, Sharrocks AD | 2023 | ChIP-seq of ADNP and TRIM28 in the U2OS cell line (Human Bone Osteosarcoma Epithelial Cells) | https://www.ebi.ac.uk/biostudies/arrayexpress/studies/E-MTAB-12292 | ArrayExpress, E-MTAB-12292 |
| Aguilar-Martinez E, Sharrocks AD | 2023 | SUMO2 ChIP-seq in the starved U2OS cells | https://www.ebi.ac.uk/biostudies/arrayexpress/studies/E-MTAB-12293 | ArrayExpress, E-MTAB-12293 |
| Aguilar-Martinez E, Sharrocks AD | 2023 | Open chromatin profiling in the U2OS cell | https://www.ebi.ac.uk/biostudies/arrayexpress/studies/E-MTAB-12294 | ArrayExpress, E-MTAB-12294 |

The following previously published datasets were used:

| Author(s) | Year | Dataset title | Dataset URL | Database and Identifier |
|---|---|---|---|---|
| Aguilar-Martinez E, Chen X, Sharrocks AD | 2015 | ZMYM2-WT, ZMYM2-SIM2mut, and FOXO3 ChIP-seq in a U2OS stable cell line | https://www.ebi.ac.uk/biostudies/arrayexpress/studies/E-MTAB-2701 | ArrayExpress, E-MTAB-2701 |
| Chen X, Ji Z, Webber A, Sharrocks AD | 2016 | Histone H3K18ac ChIP-seq in U2OS cells | https://www.ebi.ac.uk/biostudies/arrayexpress/studies/E-MTAB-3695 | ArrayExpress, E-MTAB-3695 |
| Yang F, Huang X, Zang R, Chen J | 2020 | DUX-miR-344-ZMYM2-mediated activation of MERVL LTRs induces a totipotent 2C-like state [ZMYM2 ChIP-seq] | https://www.ncbi.nlm.nih.gov/geo/query/acc.cgi?acc=GSE119818 | NCBI Gene Expression Omnibus, GSE119818 |
| Seah MKY, Wang Y, Goy PA, Loh HM | 2019 | Genome-wide binding of ZFP708/TRIM28 and linked H3K9me3 dynamics | https://www.ncbi.nlm.nih.gov/geo/query/acc.cgi?acc=GSE125673 | NCBI Gene Expression Omnibus, GSE125673 |
| Cossec JC, Theurillat I, Chica C, Búa Aguín S | 2018 | SUMO safeguards somatic and pluripotent cell identities by enforcing distinct chromatin states | https://www.ncbi.nlm.nih.gov/geo/query/acc.cgi?acc=GSE99009 | NCBI Gene Expression Omnibus, GSE99009 |
| ConsortiumRoadmap Epigenomics, Kundaje A, Meuleman W | 2015 | H3K27ac data in Human ESC H1 cells | https://egg2.wustl.edu/roadmap/data/byFileType/alignments/consolidated/E003-H3K27ac.tagAlign.gz | NIH Roadmap Epigenomics Database, E003-H3K27ac.tagAlign.gz |
| ConsortiumRoadmap Epigenomics, Kundaje A, Meuleman W | 2015 | H3K9me3 data in Human ESC H1 cells | https://egg2.wustl.edu/roadmap/data/byFileType/alignments/consolidated/E003-H3K9me3.tagAlign.gz | NIH Roadmap Epigenomics Database, E003-H3K9me3.tagAlign.gz |

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
