## [Editor Report · eLife assessment]

ZMYM2 is a transcriptional corepressor but little was known about how it is recruited to chromatin. This **important** study reveals that ZMYM2 homes to distinct classes of retrotransposons bound by the TRIM28 and ChAHP complexes in human cells, which is broadly relevant for the field of transcriptional regulation. Much of the evidence supporting the claims of the authors is **convincing**. Since widespread ZMYM2-mediated control of transposon activity is not apparent in RNA-seq data, further experiments are needed to demonstrate a more general role beyond the retrotransposons analysed in this study.

---

## [Referee Report · Reviewer #1 (Public Review)]

Owen D et al. investigated the protein partners and molecular functions of ZMYM2, a transcriptional repressor with key roles in cell identity and mutated in several human diseases, in human U2OS cells using mass spectrometry, siRNA knockdown, ChIP-seq and RNA-seq. They tried to identify chromatin bound complexes containing ZMYM2 and identified known and novel protein partners, including ADNP and the newly described partner TRIM28. Focusing mainly on these two proteins, they show that ZMYM2 physically interacts with ADNP or TRIM28, and co-occupies an overlapping set of genomic regions with ADNP and TRIM28. By generating a large set of knockdown and RNA-seq experiments, they show that ZMYM2 co-regulates a large number of genes with ADNP and TRIM28 in U2OS cells. Interestingly, ZMYM2-TRIM28 do not appear to repress genes directly at promoters, but the authors find that ZMYM2/TRIM28 repress LTR elements and suggest that this leads to gene deregulation at distance by affecting the chromatin environment within TADs.

A strength of the study is that, compared to previous studies of ZMYM2 protein partners, it investigates binding partners of ZMYM2 using the RIME method on chromatin. The RIME method makes it possible to identify low-affinity protein-protein interactions and proteins interactions occurring at chromatin, therefore revealing partners most relevant for gene regulation at chromatin. This allowed the identification of novel ZMYM2 partners not identified before, such as TRIM28.

The authors present solid interaction data with appropriate controls and generated an impressive amount of datasets (ChIP-seq for TRIM28 and ADNP, RNA-seq in ZMYM2, ADNP and TRIM28 knockdown cells) that are important to understand the molecular functions of ZMYM2. These datasets were generated with replicates and will be very useful for the scientific community. This study provides important novel insights into the molecular roles of ZMYM2 in human U2OS cells.

---

## [Referee Report · Reviewer #2 (Public Review)]

In this study the authors investigate functional associations made by transcription factor ZMYM2 with chromatin regulators, and the impact of perturbing these complexes on the transcriptome of the U2OS cell line. They focus on validating two novel chromatin-templated interactions: with TRIM28/KAP1 and with ADNP, concluding that via these distinct chromatin regulators, ZMYM2 contributes to transcriptional control of LTR and SINE retrotransposons, respectively.

Strengths of the study:

-The co-localization of ZMYM2 with ADNP and TRIM28 is validated through RIME, ChIP-seq and co-IP. Since TRIM28 is a highly abundant nuclear protein, the use of multiple methods is important to add confidence in particular for the novel SUMO-dependent interaction identified between ZMYM2 and TRIM28. That TRIM28 pulls down less of the ZMYM2-SIM mutant is reassuring.

-It is good that uniquely-mapped reads are used in the ChIP-seq analysis given the interest in repetitive elements. Likewise, though the RT-qPCR data in Fig 6 should be complemented by analysis of the RNA-seq data that the authors already have, it seems that the primers are carefully designed such that a single retrotransposon copy is amplified.

-The paper is generally written very clearly, the experiments well done and the different datasets appear to be robust.

Weaknesses of the study:

-The transcriptional response using bulk RNA-seq in ZMYM2-depleted cells remains gene-centric despite the title of the paper being about TE transcription. In fact, the only panels about TE transcription are the RT-qPCR data in Fig 6D, F. During the revision the authors said that their RNA-seq data is unfortunately too shallow to retrieve TEs. Fair enough - however, it remains the case that the central claim is control of TE transcription by ZMYM2. Thus, without additional transcriptomic analysis we are left with only a few qPCRs, even if they are nicely done! Perhaps the title could be modified a bit in that case?

-The mechanism by which ZMYM2 and TRIM28 work together does remain a mystery. Following review the authors performed TRIM28 ChIP on ZMYM2-depleted cells, but identified no changes over three transposons. It remains unclear if H3K9me3 levels are altered.

---

## [Referee Report · Reviewer #3 (Public Review)]

ZMYM2 is a transcriptional repressor known to bind to the post-translational modification SUMO2/3. It has been implicated in the silencing of genes and transposons in a variety of contexts, but lacking sequence-specific DNA binding, little is known about how it is targeted to specific regions. At least two reports indicate association with TRIM28 targets (Tsusaka 2020 Epigenetics & Chromatin, Graham-Paquin 2023 NAR) but no physical association with TRIM28 targets had been demonstrated. Tsusaka 2020 theorizes an indirect, potentially SUMO-independent, interaction via ATF7IP and SETDB1.

Here, Owen and colleagues show that a subset of ZMYM2-binding sites in U2OS cells are clearly TRIM28 sites, and further find that hundreds of genes are silenced by both ZMYM2 and TRIM28. They next demonstrate that ZMYM2 homes to chromatin, and interacts with TRIM28, in a SUMOylation-dependent manner, suggesting that ZMYM2 is recognizing SUMOylation on TRIM28 or a protein associated with TRIM28. ZMYM2 separately homes to SINE elements bound by the ChAHP complex in an apparently SUMOylation independent manner. Although this is not the first report to show physical interaction between ZMYM2 and ChAHP, it is the first to show that ZMYM2 homes to ChAHP-binding sites and functions as a corepressor at these sites. Finally the authors demonstrate that ZMYM2 and TRIM28 coregulate genic targets by inducing repression at LTRs within the same TADs as the genes in question.

Overall, the manuscript is well-written, convincing, and fills a significant hole in our understanding of ZMYM2's mechanistic function. The revised version of this manuscript addresses all of my previous concerns well.

---

## [Author Response]

The following is the authors’ response to the original reviews.

Thank you for your recent editorial decision on our manuscript. I have included a revised version of our manuscript in which we have addressed all of the required editorial and referees’ comments as requested. In summary, we have added substantial amounts of new data and analysis (new Fig. 5D; Supplementary Figures S1E, S3C, S3E, S3I, S4C), amended several figures (Figures 2 and 3), added a new supplementary Table (Table S2) and we have changed the text and figure labelling/presentation in appropriate places to clarify or correct the issues raised by the reviewers.

In summary, we firmly believe that we have addressed all the outstanding issues in a positive manner and that the manuscript is now suitable for publication in eLife. I look forward to receiving your final editorial decision on this manuscript.

**eLife assessment:**
ZMYM2 is a transcriptional corepressor but little was known about how it is recruited to chromatin. This study reveals that ZMYM2 homes to distinct classes of retrotransposons bound by the TRIM28 and ChAHP complexes in human cells, an important finding in the field of transcriptional regulation. The evidence supporting the claims of the authors is solid, although inclusion of more functional data would have strengthened the original model proposed.

We have taken all the comments on board and provided additional new experimental data where requested and more data analysis to substantiate our claims.

**Reviewer #1 (Public Review):**
Owen D et al. investigated the protein partners and molecular functions of ZMYM2, a transcriptional repressor with key roles in cell identity and mutated in several human diseases, in human U2OS cells using mass spectrometry, siRNA knockdown, ChIP-seq and RNA-seq. They tried to identify chromatin bound complexes containing ZMYM2 and identified known and novel protein partners, including ADNP and the newly described partner TRIM28. Focusing mainly on these two proteins, they show that ZMYM2 physically interacts with ADNP or TRIM28, and co-occupies an overlapping set of genomic regions with ADNP and TRIM28. By generating a large set of knockdown and RNA-seq experiments, they show that ZMYM2 co-regulates a large number of genes with ADNP and TRIM28 in U2OS cells. Interestingly, ZMYM2-TRIM28 do not appear to repress genes directly at promoters, but the authors find that ZMYM2/TRIM28 repress LTR elements and suggest that this leads to gene deregulation at distance by affecting the chromatin environment within TADs.A strength of the study is that, compared to previous studies of ZMYM2 protein partners, it investigates binding partners of ZMYM2 using the RIME method on chromatin. The RIME method makes it possible to identify low-affinity protein-protein interactions and proteins interactions occurring at chromatin, therefore revealing partners most relevant for gene regulation at chromatin. This allowed the identification of novel ZMYM2 partners not identified before, such as TRIM28.The authors present solid interaction data with appropriate controls and generated an impressive amount of datasets (ChIP-seq for TRIM28 and ADNP, RNA-seq in ZMYM2, ADNP and TRIM28 knockdown cells) that are important to understand the molecular functions of ZMYM2. These datasets were generated with replicates and will be very useful for the scientific community. This study provides important novel insights into the molecular roles of ZMYM2 in human U2OS cells.The authors could have been more precise in the manuscript title and abstract to emphasize that these findings apply to human cells, as indeed there is no demonstration yet that the findings presented here can be transposed to mouse cells.

We have slightly changed the title and abstract to emphasise that the findings are in human cells.

The manuscript's main conceptual advance is that the authors propose a novel model of gene regulation whereby transcriptional repressors of transposable elements could regulate genes at distance by modulating the local chromatin environment within TADs. Additional experiments would be needed to strengthen this model. For example the authors could have performed TRIM28 ChIP in ZMYM2-kd cells to test if ZMYM2 favors the recruitment of TRIM28 to its genomic targets, as well as ChIP-seq of repressive chromatin marks (such as H3K9me3) in ZMYM2-kd cells to investigate if the loss of ZMYM2 leads to reduced H3K9me3 in ERVs and over large regions surrounding the ERVs.

We have tested whether ZMYM2 is required for TRIM28 binding at several loci and find no evidence for this (new Supplementary Fig. S3E). We now discuss this in the results text and discussion where we already suggested that TRIM28 is likely recruited by KRAB-zinc finger proteins and ZMYM2 is subsequently recruited to this complex. Future extensive work is required to understand the mechanistic functions of ZMYM2 in these regions.

**Reviewer #2 (Public Review):**
In this study the authors investigate functional associations made by transcription factor ZMYM2 with chromatin regulators, and the impact of perturbing these complexes on the transcriptome of the U2OS cell line. They focus on validating two novel chromatin-templated interactions: with TRIM28/KAP1 and with ADNP, concluding that via these distinct chromatin regulators, ZMYM2 contributes to transcriptional control of LTR and SINE retrotransposons, respectively.Strengths and weakness of the study:The co-localization of ZMYM2 with ADNP and TRIM28 is validated through RIME, ChIP-seq and co-IP. (Notably, since both RIME and ChIP-seq rely on crosslinking, and the co-IP with TRIM28 required crosslinking due to being SUMO-dependent, only the ZMYM2-ADNP co-IP experiment demonstrates an interaction in the absence of crosslinking).

This is not correct as the co-IP experiments between endogenous ZMYM2 and TRIM28 were not performed in the presence of cross linkers. They did have NEM added, but this was to inactivate SUMO proteases rather than to cross link proteins.

It is good that uniquely-mapped reads are used in the ChIP-seq analysis given the interest in repetitive elements. Likewise, though the RT-qPCR data in Fig5 should be complemented by analysis of the RNA-seq data that the authors already have, it seems that the primers are carefully designed such that a single retrotransposon copy is amplified.

We re-analysed our RNA-seq data using the TEtranscripts tool and looked at TE transcription genome-wide. However very few TEs were expressed at high enough levels to get any statistically significant additional data beyond a few additional transposable elements. This likely results from the relatively low read depth we used and the lack of specific protocols being followed to preserve TE transcripts. We will return to the genome-wide effects in future studies where we plan to switch cell types and will generate more bespoke datasets (the current ones were designed for analysing effects on protein coding gene expression before we made the connection to TEs). We added additional text to the results section to indicate that we could not see widespread deregulation of subclasses of TEs but that this needs further work.

The top-scoring interactors are highly-abundant nuclear proteins: for example, data from the contaminant repository for affinity purification mass-spec data (https://reprint-apms.org/) show that TRIM28 is identified in 466 / 716 AP-MS experiments with a mean spectral count of 16. While this does not indicate that the ZMYM2-TRIM28 interaction is not 'true', it would have been helpful to further dissect the interaction to strengthen this conclusion. For example, it would be nice to see the co-IP (fig 3A) repeated from the cells expressing the ZMYM2 mutant that is no longer competent to bind SUMO (used in the ChIP-seq data of Fig 2). Alternatively - if the model is that ZMYM2 recruits SUMOylated TRIM28 with well-characterized TRIM28 mutants that lack SUMOylation.

We are aware that TRIM28 is often present as an apparent contaminant in many mass spec studies. However we have provided co-IP, PLA and ChIP-seq data to support their co-association on chromatin. We also convincingly show that ZMYM2 and TRIM28 functionally converge on regulating the same gene expression programmes. As requested by the referee, we have added further data showing that the ZMYM2 protein that is defective in SUMO binding (ZMYM2(SIM2mut); new Supplementary Fig. S3C) shows reduced binding to TRIM28 in co-IP assays. This further strengthens the (SUMO-dependent) association between ZMYM2 and TRIM28.

The transcriptional response using bulk RNA-seq in ZMYM2-depleted cells is rather gene-centric despite the title of the paper being about TE transcription. In fact the only panels about TE transcription are the RT-qPCR data in Fig 5D,F. I may be missing something (and there aren't many details given about the RNA-seq experiments) but why not look at TE transcription in an unbiased way with the transcriptomic data at hand? I appreciate potential hazards of multi-mapping etc but it would be interesting to see at least some subfamily analysis (e.g. using the TEtranscripts tool). On a similar point, why not show some RNA-seq in the genome browser snapshots of the epigenomics - together with a RepeatMasker annotation track of TEs...

See response to the same point above.

While the results broadly support the authors' conclusions, I have the overall impression that the central claim of TE transcriptional regulation by ZMYM2 could be strengthened a lot with some fairly straightforward additional experiments and analyses.
**Reviewer #3 (Public Review):**
ZMYM2 is a transcriptional repressor known to bind to the post-translational modification SUMO2/3. It has been implicated in the silencing of genes and transposons in a variety of contexts, but lacking sequence-specific DNA binding, little is known about how it is targeted to specific regions. At least two reports indicate association with TRIM28 targets (Tsusaka 2020 Epigenetics & Chromatin, Graham-Paquin 2022 bioRxiv) but no physical association with TRIM28 targets had been observed. Tsusaka 2020 theorizes an indirect, potentially SUMO-independent, interaction via ATF7IP and SETDB1.Here, Owen and colleagues show that a subset of ZMYM2-binding sites in U2OS cells are clearly TRIM28 sites, and further find that hundreds of genes are silenced by both ZMYM2 and TRIM28. They next demonstrate that ZMYM2 homes to chromatin, and interacts with TRIM28, in a SUMOylation-dependent manner, suggesting that ZMYM2 is recognizing SUMOylation on TRIM28 itself. ZMYM2 separately homes to SINE elements bound by the ChAHP complex, in an apparently SUMOylation independent manner. Although this is not the first report to show physical interaction between ZMYM2 and ChAHP, it is the first to show that ZMYM2 homes to ChAHP-binding sites and functions as a corepressor at these sites.The mode by which ZMYM2 and TRIM28 coregulate genic targets remains somewhat unclear. TRIM28/ZMYM2 bind to LTR elements, loss of these proteins results in upregulation of genes distal to (but in the same TAD as) these binding sites.Overall, the manuscript is well-written, convincing, and fills a significant hole in our understanding of ZMYM2's mechanistic function.

We thank the referee for his/her positive evaluation of the mechanistic insights we provide. We have further added to these through addressing the specific issues raised in their “recommendations for authors”.

**Recommendations for the authors:**

The reviewers appreciated the novelty of the findings, and in particular, the use of the RIME method to identify the protein partners of ZMYM2 while bound on chromatin, and multiple validation steps of these novel ZMYM2 interactors. However, they also felt that the model presented at the end of the manuscript seems preliminary and would deserve additional experiments to be really supported, the essential ones being listed below:

1 - Despite the claimed scope of the manuscript on TE regulation, their expression analysis is limited to RT-qPCR and targeted to a few families or copies. Please use the RNA-seq data generated in U2OS cells depleted for ZMYM2 to assess retrotransposon expression genome-wide, performing both family-level and copy-level analyses, and compare with TRIM28-depleted U2OS cells.

We re-analysed our RNA-seq data using the TEtranscripts tool and looked at TE transcription genome-wide. However very few TEs were expressed at high enough levels to get any statistically significant additional data beyond a few additional transposable elements. This likely results from the relatively low read depth we used and the lack of specific protocols being followed to preserve TE transcripts. We will return to the genome-wide effects in future studies where we plan to switch cell types and will generate more bespoke datasets (the current ones were designed for analysing effects on protein coding gene expression before we made the connection to TEs). We added additional text to the results section to indicate that we could not see widespread deregulation of subclasses of TEs but that this needs further work.

2 - Clarify the relationship between dysregulated genes and TAD boundaries, as this seems important to support the model of distant gene regulation by the action of ZMYM2 on local chromatin environment within TADs (see comment of Reviewer #1 and #3).

We have now provided further support for the idea that ZMYM2 functions within TADs as detailed below in response to the reviewers comments. New bioinformatics analysis has been done which is incorporated into the paper in Fig. 4D and Supplementary Fig. S4C.

3 - Perform TRIM28 ChIP-seq in ZMYM2-kd cells, to prove that ZMYM2 indeed participates to TRIM28 recruitment to TE loci. This could be complemented by H3K9me3 ChIP-seq, to see if ZMYM2 depletion reduces H3K9me3 at retroytransposons, and over the regions surrounding ERVs. This last experiment seems also important for reinforcing the distant regulation model of nearby genes through ZMYM2-mediated repression of retrotransposons.

As suggested by the referees below, we have tested whether ZMYM2 is required for TRIM28 binding at several loci and find no evidence for this (new Supplementary Fig. S3E). We now discuss this in the results text and discussion where we already suggested that TRIM28 is likely recruited by KRAB-zinc finger proteins and ZMYM2 is subsequently recruited to this complex. Future extensive work is required to understand the mechanistic functions of ZMYM2 in these regions.

**Reviewer #1 (Recommendations For The Authors):**
Figure S1D is not clear. The authors want to investigate if ADNP and ZMYM2 regulate gene expression in the same directionality. They compare the genes down in siADNP and up in siZMYM2 (or vice versa) and show very small overlaps. If I understand correctly, this shows that very few genes are regulated in opposite directions by ADNP and ZMYM2 and consequently that they tend to regulate genes in the same directionality. This is not what is said in the text page 19 ("with no clear common roles as either an activator or repressor") and should be clarified. Furthermore, to compare if ADNP and ZMYM2 regulate genes in the same directionality, there are better ways to represent this, for example scatter plots of log2 FC in ADNP kd vs ZMYM2 kd. Similar criticisms apply to Fig S3F.

We agree that the text could be clearer and have rewritten it as “….although the large numbers of genes directionally co-regulated by these two proteins (ie either positively or negatively) indicates no clear common role as either an activator or repressor”. We have also added a scatter plot to the supplementary data (Fig. S1E) to further emphasise the common directionality of effect as suggested by the reviewer. Similarly, we changed the text and have added a scatter plot to support the conclusions on ZMYM2 and TRIM28 functional interactions (new Fig. S3I).

The authors suggest an indirect control of genes by ZMYM2 within TADs (Fig 4C). Yet Fig 4C does not seem to address this point. Fig 4C shows that TADs with a ZMYM2/cluster 1 peak contain more upregulated than downregulated genes, but the key question should be: are upregulated genes significantly enriched in TADs containing a ZMYM2/cluster 1 peak compared to other TADs or other genomic regions?

We have taken this suggestion on board and determined the frequency distribution of the number of TADs containing a gene upregulated (fold change >1.6; Padj <0.01) following ZMYM2 depletion. 10,000 iterations were performed by randomly selecting 216 TADs across all 3062 TADs. The observed number of TADs containing an upregulated gene (42) from 216 TADs containing a cluster 1 ZMYM2 peak is a clear outlier in this distribution (P-value = 0.0002) (see Supplementary Fig. S4C).

A key question not addressed in the manuscript is whether ZMYM2 participates in the recruitment of TRIM28 to ERVs. I recommend performing TRIM28 ChIP in ZMYM2-kd cells.

We have tested whether ZMYM2 is required for TRIM28 binding at several loci and find no evidence for this (new Supplementary Fig. S3E). We now discuss this in the results text and discussion where we already suggested that TRIM28 is likely recruited by KRAB-zinc finger proteins and ZMYM2 is subsequently recruited to this complex. Future extensive work is required to understand the mechanistic functions of ZMYM2 in these regions.

**Reviewer #2 (Recommendations For The Authors):**
Please give more details of RNA-seq analyses in the experimental section (this will be particularly important if the comment about analysing TE transcription genome-wide is acted on).

We have now expanded on the description of the RNA-seq analysis including adding in the mapping statistics to a new Supplementary table. We followed the referee’s useful suggestion of looking at TE transcription genome-wide. However very few TEs were expressed at high enough levels to get any statistically significant additional data. This likely results from the relatively low read depth we used and the lack of specific protocols being followed to preserve TE transcripts. We will return to the genome-wide effects in future studies where we plan to switch cell types and will generate more bespoke datasets (the current ones were designed for analysing effects on protein coding gene expression before we made the connection to TEs).

**Reviewer #3 (Recommendations For The Authors):**
Major Comments:The relationship of TRIM28/ZMYM2 repression of LTRs and silencing within/between TADs is interesting but underdeveloped. Upon ZMYM2 depletion, the authors observe simultaneous upregulation of genes within TADs more often than would be expected by chance, but this analysis does not distinguish "proximal to" from "in the same TAD". If a ZMYM2 binding site is X bases from a gene TSS, is it more likely to regulate that gene if it is in the same TAD? This can and should be tested bioinformatically.

The basic question the referee is asking is whether ZMYM2 affects gene expression at a certain distance irrespective of whether the TSS of the gene is in the same TAD. We have now tested this and added text to the results section. Basically we took all of the ZMYM2 regions associated with genes upregulated by ZMYM2 depletion that resided in the same TAD and calculated the peak to TSS distance. Then we searched in the opposite direction for the TSS of genes at a similar distance (+/-25%) that resided in an adjacent TAD. We then asked whether these genes were upregulated by ZMYM2 depletion. 102 ZMYM2 peaks were positioned within these distance constraints with at least one gene in an adjacent TAD (716 genes in total). Of these genes, only 11 were upregulated following ZMYM2 depletion. There is therefore not a general spreading of deregulation around ZMYM2 peaks in a distance-dependent manner.

Furthermore, the authors note in the text and discussion that LTRs can demarkate TAD boundaries, but this is a distinct concept from the idea that they regulate genes within a TAD. Is there evidence that ZMYM2 binding sites are found at TAD boundaries?

We have provided more evidence to support the associations of ZMYM2 peaks with TADs and now show that they are closer than randomly expected to TAD boundaries (Fig. 4D). However they are clearly not all located very close to the boundaries.

The analysis of transposons expression was limited to qPCR of a handful of elements. Since the authors have conducted RNA-seq of U2OS cells depleted for both TRIM28 and ZMYM2, they can determine if certain classes of transposons are globally upregulated.

We re-analysed our RNA-seq data using the TEtranscripts tool and looked at TE transcription genome-wide. However very few TEs were expressed at high enough levels to get any statistically significant additional data. This likely results from the relatively low read depth we used and the lack of specific protocols being followed to preserve TE transcripts. We will return to the genome-wide effects in future studies where we plan to switch cell types and will generate more bespoke datasets (the current ones were designed for analysing effects on protein coding gene expression before we made the connection to TEs). We added additional text to the results section to indicate that we could not see widespread deregulation of subclasses of TEs but that this needs further work.

Minor Comments:Typo: "human HEK393 cells". They are HEK293 cells.

We have corrected this error.

"These ADNP peaks showed enrichment of binding motifs for several transcription factors with the top two motifs for HBP1 and IRF both found in over 35% of target regions (Figure 1D)." According to Ostapcuz 2018, ADNP has its own motif (CGCCCYCTNSTG). It is intriguing that this does not appear enriched in ADNP sites in U2OS cells, this seems worthy of comment.

This is a good point, so we did an additional search using the motif found in Ostapcuk 2018 and found this in 15% of ADNP binding regions. This value is substantially lower than the 63% seen previously. It therefore is present but is not the dominant motif. This new data and its implication regarding chromatin targeting mechanisms is now discussed in the Results section around Fig. 1D.

Figures S2F and S2G are central to the paper and belong in the main text.

We have now added these to the main figures as requested (meaning that Fig.2 has now been split into two separate figures {2 and 3} as became too large for a single figure).

A supplementary table including libraries generated and mapping statistics should be included.

We have now added this (new Supplementary Table S2)